# Evapotranspiration in the Amazon: spatial patterns, seasonality and recent trends in observations, reanalysis and CMIP models

Jessica C.A. Baker[1], Luis Garcia-Carreras[2], Manuel Gloor[3], John H. Marsham[1,4], Wolfgang Buermann[5], Humberto R. da Rocha[6], Antonio D. Nobre[7], Alessandro Carioca de Araujo[8] and Dominick V. Spracklen[1]

[1]School of Earth and Environment, University of Leeds, Leeds, UK
[2]School of Earth and Environmental Sciences, University of Manchester, UK
[3]School of Geography, University of Leeds, Leeds, UK
[4]National Centre for Atmospheric Science, Leeds, UK
[5]Institut fuer Geographie, Universitaet Augsburg, D-86135 Augsburg, Germany
[6]Departamento de Ciências Atmosféricas, Instituto de Astronomia, Geofísica e Ciências Atmosféricas, Universidade de São Paulo, Brasil
[7]Earth System Science Center, INPE, Av dos Astronautas, 1758, São José dos Campos, SP Brasil
[8]Empresa Brasileira de Pesquisa Agropecuária (EMBRAPA), Belém-PA, CEP 66095-100, Brasil

*Correspondence to*: Jessica C.A. Baker (J.C.Baker@Leeds.ac.uk)

**Abstract.** Water recycled through transpiring forests influences the spatial distribution of precipitation in the Amazon and has been shown to play a role in the initiation of the wet season. However, due to the challenges and costs associated with measuring evapotranspiration (ET) directly, plus the high uncertainty and discrepancies across remote-sensing retrievals of ET, spatial and temporal patterns in Amazon ET remain poorly understood. In this study, we estimated ET over the Amazon and ten sub-basins using a catchment-balance approach, whereby ET is calculated directly as the balance between precipitation, runoff and change in groundwater storage. We compared our results with ET from remote-sensing datasets, reanalysis, models from the fifth and sixth Coupled Model Intercomparison Projects (CMIP5 and CMIP6), and in-situ flux-tower measurements to provide a comprehensive overview of current understanding. Catchment-balance analysis revealed a gradient in ET from east to west/southwest across the Amazon basin, a strong seasonal cycle in basin-mean ET primarily controlled by net incoming radiation, and no trend in ET over the past two decades. This approach has a degree of uncertainty, due to errors in each of the terms of the water-budget, therefore we conducted an error analysis to identify the range of likely values. Satellite datasets, reanalysis and climate models all tended to overestimate the magnitude of ET relative to catchment-balance estimates, underestimate seasonal and interannual variability, and show conflicting positive and negative trends. Only two out of six satellite and model datasets analysed reproduced spatial and seasonal variation in Amazon ET, and captured the same controls on ET as indicated by catchment-balance analysis. CMIP5 and CMIP6 ET was inconsistent with catchment-balance estimates over all scales analysed. Overall, the discrepancies between data products and models revealed by our analysis demonstrate a need for more ground-based ET measurements in the Amazon, and to substantially improve model representation of this fundamental component of the Amazon hydrological cycle.

**Keywords**: transpiration; tropical forest; water-budget; remote sensing; CMIP6

## 1 Introduction

Evapotranspiration (ET) is the transfer of water from the land to the atmosphere through evaporation from soil, open water and canopy-intercepted rainfall, plus transpiration from plants. More than half of all water that falls as precipitation over land is recycled back to the atmosphere through ET (Schlesinger and Jasechko, 2014, Good et al., 2015, Jasechko, 2018). This essential hydrological process affects the partitioning of heat fluxes at the Earth's surface, causing local cooling, while providing moisture for precipitation, thus sustaining the hydrological cycle (Jung et al., 2010, Wang and Dickinson, 2012, Zhang et al., 2016a). Transpiration is the dominant component of terrestrial ET, and transpiration rates over tropical forests are among the highest in the world (Zhang et al., 2001, Jasechko et al., 2013, Good et al., 2015, Wei et al., 2017). In the Amazon, where tropical forest covers approximately 5.5 M km$^2$, sap flux measurements from a site near Manaus showed the transpiration contribution to ET increased from 40 % in the wet season up to 95 % in the driest part of the year (Kunert et al., 2017).

Amazon ET is essential for maintaining the regional hydrological cycle and sustaining a climate favourable for tropical rainforests (Salati and Vose, 1984, Eltahir and Bras, 1994, Eltahir, 1996, Nepstad et al., 2008, van der Ent et al., 2010, Zemp et al., 2014). Consequently, changes in ET have implications for local and regional climate (Spracklen et al., 2012, Silvério et al., 2015, Spracklen et al., 2018, Baker and Spracklen, 2019) and may impact the stability of the Amazon forest biome (Zemp et al., 2017b). Deforestation, which has seen a recent upsurge in the region (Barlow et al., 2020), causes reductions in ET, though the magnitude of the response is still not fully understood. Estimates based on in-situ and remote-sensing data from the southern Amazon suggest deforestation-driven ET reductions range from 15 to 40% in the dry season (von Randow et al., 2004, Da Rocha et al., 2009b, Khand et al., 2017, da Silva et al., 2019). Changes in the global climate are also affecting Amazon ET by increasing atmospheric demand for water vapour, resulting in positive ET trends since the 1980s (Zhang et al., 2015a, Zhang et al., 2016b, Pan et al., 2020). Over the next century, coupled climate models suggest there may be large reductions in ET as plants reduce stomatal conductance in response to rising atmospheric $CO_2$ (Skinner et al., 2017, Kooperman et al., 2018), leading to changes in the surface energy balance and atmospheric circulation that drive reductions in Amazon rainfall (Langenbrunner et al., 2019). To assess changes in ET over the Amazon and evaluate climate model credibility, reliable observations of ET are required. However, despite being integral to the health of the Amazon ecosystem, ET over this region remains a challenging variable to measure and quantify (Pan et al., 2020).

Several early studies used measurements of stable water isotopes to evaluate water recycling in the Amazon, since the isotope composition of transpired water is distinct from that of evaporated water (Salati et al., 1979, Victoria et al., 1991, Martinelli et al., 1996, Moreira et al., 1997). Such work first highlighted the predominance of transpiration over the Amazon, relative to continental areas with lower forest cover, such as Europe (Salati et al., 1979, Gat and Matsui, 1991). More recently, studies based on satellite retrievals of hydrogen isotopes in tropospheric water vapour have suggested that transpiration could be key

in triggering convection during the Amazon dry-to-wet season transition (Wright et al., 2017), and that ET reductions in the 2005 drought caused a delay in wet-season onset in the following year (Shi et al., 2019). However, while isotopes can help to partition ET into its respective components, they cannot provide information about the absolute magnitude of the ET flux, and thus other methods are required to quantify ET.

Amazon ET can be quantified using a catchment-balance approach, whereby ET is approximated as the difference between precipitation and runoff. Estimates of annual mean Amazon ET using this method range from 992 to 1905 mm yr$^{-1}$ (mean±σ = 1421±254 mm yr$^{-1}$; Marengo, 2006 and references therein), though part of this uncertainty is due to differences in the definition of Amazon basin extent. Historically, water budget approaches have assumed that groundwater storage does not change over time, though more recent studies have been able to also account for changes in groundwater using terrestrial water storage anomalies measured by the Gravity Recovery and Climate Experiment (GRACE) satellites (i.e. Swann and Koven, 2017, Maeda et al., 2017, Sun et al., 2019). Swann and Koven (2017) estimated annual mean Amazon ET to be 1058 mm yr$^{-1}$, which is towards the lower end of previous estimates. Constraining Amazon ET in this way is useful, though a whole-basin-scale analysis by definition does not capture spatial variation in Amazon ET. Maeda et al. (2017) used a water balance approach to estimate ET in five Amazon sub-basins, and found values ranging from 986 mm yr$^{-1}$ in the Solimões basin in the western Amazon, to 1497 mm yr$^{-1}$ in the northern Negro basin. However, even this sub-basin-scale analysis is likely to mask finer-scale spatial heterogeneities in ET.

Direct, site-level measurements of ET can be obtained from eddy-covariance (EC) flux towers. During the 1990s, a network of towers was established in Brazil as part of the Large-Scale Biosphere-Atmosphere Experiment in Amazonia (LBA) research programme (see Keller et al., 2009, and references therein). ET measurements from these towers have provided valuable insights into the drivers of variability in Amazon ET and how ET varies over different temporal scales (da Rocha et al., 2004, Hasler and Avissar, 2007, Fisher et al., 2009, Restrepo-Coupe et al., 2013, Christoffersen et al., 2014). EC data have shown that surface net radiation is the primary control on seasonal Amazon ET over wet areas of the Amazon (precipitation above 1900 mm), while variation in water availability governs ET in the seasonally-dry tropical forests in the south and southeast Amazon, towards the boundary with the Cerrado biome (da Rocha et al., 2009a, Costa et al., 2010). Despite these advances in understanding, it should be noted that EC measurements have an inherent degree of uncertainty, as measured turbulent heat fluxes do not sum to the total measured available energy (i.e. the energy balance closure problem, Wilson et al., 2002, Foken, 2008). Tropical forest LBA tower sites underestimated the total energy flux by 20–30 % (Fisher et al., 2009), indicating that part of the ET flux might have been missed. A study in western Europe also suggested that flux towers may underestimate ET over forests compared to ET from lysimeters and water-balance methods (Teuling, 2018). Variation in energy closure between flux tower sites also makes it difficult to make direct comparisons between absolute ET values measured in different locations, presenting a further challenge (da Rocha et al., 2009a). Finally, the spatial distribution of flux towers in South America is uneven, with no EC ET measurements currently available over large areas of the western and northern Amazon (see Fig. 1).

Given the relatively high costs associated with setting up and running flux towers, and the inaccessibility of much of the Amazon basin, it is desirable to find alternative methods of monitoring ET over this region of remote tropical forest and elsewhere.

Over the past few decades, ET products derived from Earth observation satellites have become available (e.g. Mu et al., 2007, Zhang et al., 2010, Miralles et al., 2011, Mu et al., 2011, Martens et al., 2017). These products offer ET estimates over previously unmonitored regions, such as the western Amazon, and therefore have potential to further our understanding of the controls and drivers of the Amazon hydrological cycle. Satellite-based ET products provide spatially and temporally

homogeneous information at scales that are well suited for climate model evaluation. However, it is important to note that these products are not direct measures of ET, but rather ET is estimated from variables that satellites do measure (essentially radiation), other satellite retrievals (e.g. leaf area index, LAI) and, crucially, model-derived inputs. Thus, although often referred to as 'observational datasets', it is more accurate to consider satellite ET products as physically-constrained land-surface models. Global-scale ET product comparisons have been conducted before, for example as part of the WACMOS-ET

(WAter Cycle Multi-mission Observation Strategy – EvapoTranspiration) project (Michel et al., 2016, Miralles et al., 2016), and a more recent detailed evaluation that included multiple remote-sensing datasets and 14 land-surface models (Pan et al., 2020). While these studies made some comparisons between products over the Amazon, they did not include any 'ground-truth' validation data over South America. Further work has evaluated satellite ET products over the Amazon at different spatial scales (e.g. Ruhoff et al., 2013, Maeda et al., 2017, Swann and Koven, 2017, de Oliveira et al., 2017, Sörensson and

Ruscica, 2018, Paca et al., 2019, Wu et al., 2020), though a detailed analysis of spatial and temporal variation in remote-sensing ET products, evaluated against ET from catchment-balance analysis and flux towers, is currently lacking.

Finally, representation of Amazon ET in coupled climate models is still underdeveloped, in part due to limited high-quality reference observations. To overcome uncertainties in benchmarking data, Mueller and Seneviratne (2014) utilised a synthesis

of 40 observational, reanalysis and land-surface model datasets (Mueller et al., 2013) to evaluate 14 models from the fifth Coupled Model Inter-comparison Project (CMIP5). Their analysis showed that Amazon ET tended to be overestimated at the annual scale, but underestimated from June to August. More recently it was observed that 28 out of 40 CMIP5 models misrepresented the controls on Amazon ET, with implications for future precipitation projections in the region (Baker et al., in review). Other assessments of CMIP5 models over the Amazon have found that choice of reference ET dataset can have a

large impact on model performance metrics (Schwalm et al., 2013, Baker et al., 2021). Catchment-balance analysis accounting for changes in groundwater storage, offers an alternative approach for directly quantifying Amazon ET and its associated uncertainty at the monthly timescale, but to our knowledge has not previously been applied to evaluate climate models. With output from the sixth generation of CMIP models now available (Eyring et al., 2016), there is an opportunity to extend earlier evaluation studies by comparing simulated Amazon ET against catchment-balance estimates, and thus provide a first

assessment of model performance over the Amazon.

The aim of this study was to summarise the current 'state-of-the-science' for Amazon ET in an attempt to determine what aspects of Amazon ET are well understood, identify areas of remaining uncertainty, and provide a benchmark to evaluate the latest generation of coupled climate models. Given the challenges associated with estimating ET, we collated data from a variety of sources, expanding earlier studies by including 'direct' estimates of ET from catchment-balance analysis and flux towers in our validation, and deriving ET estimates for ten Amazon sub-basins, permitting an assessment of controls on spatial variation in ET. Our results highlight substantial differences between ET products, while our catchment-balance analysis provides new insights on the spatial and temporal patterns of ET variability over the Amazon basin.

## 2 Data and Methods

To capture a complete spectrum of ET estimates over the Amazon, we combined data from catchment-balance analysis, flux towers, remote-sensing products, reanalysis and coupled climate models. The origins of these datasets are described in the sections that follow.

### 2.1 Catchment-balance ET

Catchment-balance ET provides the closest approximation to a direct ET 'measurement' over large spatial scales in this study. In this approach, ET is calculated as the difference between terms in the water-budget equation that can be measured (within a margin of error), following Eq. (1):

$$\text{ET} = P - R - \frac{dS}{dt} \qquad (1)$$

where $P$ is area-weighted, catchment-mean precipitation, $R$ is river runoff from the basin, and $\frac{dS}{dt}$ is the area-weighted, basin-mean change in terrestrial water storage ($S$) over the basin with respect to time ($t$), all in units of mm month$^{-1}$. Catchment-balance ET was calculated, first as the simple difference between precipitation and runoff (climatological basin means only), and then using the more sophisticated approach that accounts for temporal variation in groundwater storage (Rodell et al., 2011, Long et al., 2014, Swann and Koven, 2017, Maeda et al., 2017, Sun et al., 2019).

The catchment-balance approach was used to estimate climatological annual mean ET for the Amazon basin and ten sub-catchments: the Solimões, Japura, Negro, Branco, Jari, Purus, Madeira, Aripuanã, Tapajos and Xingu catchments (Fig. 1). Temporal variation in catchment ET was analysed for the Amazon basin only. Basin domains were constructed by aggregating sub-basin shapefiles that had previously been identified using a digital elevation model (Seyler et al., 2009), making sure to include all sub-basins upstream of the relevant river station.

Precipitation data came from the 0.05° x 0.05° Climate Hazards Group InfraRed Precipitation with Station (CHIRPS) version 2.0 dataset, which combines data from satellites and rain gauges (Funk et al., 2015). CHIRPS has been validated against rain-gauge data from northeast Brazil, including four Amazon stations, and found to have mean bias and absolute error values of –3.6% and 28.4 mm month$^{-1}$, respectively (Paredes-Trejo et al., 2017).

Monthly-mean river flow data were obtained from the Agência Nacional de Águas (ANA) database in Brazil (HidroWeb, 2018). To obtain runoff in mm month$^{-1}$, volumetric flow rates (m$^3$ s$^{-1}$) were divided by the catchment area (m$^2$), scaled to the monthly timestep by multiplying by the number of seconds in each month and multiplied by 1000 to convert to mm. To estimate 'whole' Amazon ET, we used runoff measured at Óbidos, which drains approximately 77 % (Callède et al., 2008) of the Amazon basin (Fig 1). For the Tapajos catchment, runoff from Itaituba, was gap-filled based on linear regression with data
from the Buburé station, which is approximately 70 km upstream (R$^2$=0.77, 15 data points in total). Details of the gauge stations used for the other basin river records are provided in Table S1.

Terrestrial water storage data were derived from the 0.5° x 0.5° Jet Propulsion Laboratory (JPL) RL06M Version 2.0 GRACE Mascon Solution, with Coastline Resolution Improvement (CRI) filtering and land-grid scaling factors (derived from the
Community Land Model, CLM) applied (Watkins et al., 2015, Wiese et al., 2016, Wiese et al., 2018). This dataset, which has been processed to minimise measurement errors and optimise the signal-to-noise ratio, represents a new generation of GRACE solutions that do not require empirical post-processing to remove correlated errors, and are thus considered more rigorous than the previous GRACE land water storage estimates based on spherical-harmonic solutions (Wiese et al., 2016).

To determine the change in water storage, $\frac{dS}{dt}$, in units of mm month$^{-1}$, we calculated the difference between consecutive GRACE measurements for each grid cell, divided by the time between measurements, as shown in Eq. (2):

$$\frac{dS}{dt} = (S_{[n]} - S_{[n-1]})/dt \qquad (2)$$

where $S$ represents the land water storage anomaly in mm, $n$ is the measurement number and $dt$ is the time between measurements $[n]$ and $[n–1]$ in months. Following this, we calculated the area-weighted, basin-mean $\frac{dS}{dt}$ for each catchment.
Finally, to account for the uneven temporal sampling of GRACE data (due to battery management on the GRACE satellites), we used a linear spline to interpolate $\frac{dS}{dt}$ values to the same temporal grid as the precipitation and runoff data, i.e. one value per month for the period May 2002 to December 2019.

Previous work has shown that GRACE is less sensitive at lower latitudes than at higher latitudes, and may only be capable of
detecting monthly changes in groundwater storage over regions larger than 200,000 km$^2$, or seasonal changes over areas greater than 184,000 km$^2$ (Rodell and Famiglietti, 1999). Three of the basins included in this analysis have areas smaller than these

thresholds, namely Jari (49,000 km$^2$), Branco (131,000 km$^2$) and Aripuanã (138,000 km$^2$, Table S1). However, we only computed climatological means over these basins, and the catchment-balance ET estimates were in excellent agreement with ET calculated as the difference between precipitation and runoff (r=0.997, p<0.001, Fig. S1). Therefore, we have confidence

that our results for these basins were not biased by the inclusion of GRACE in the calculations.

For the Amazon basin only, we calculated catchment-balance ET at the monthly timescale. We estimated the relative uncertainty of our ET estimates ($\upsilon_{ET}$) by propagating errors in each of the terms of the water budget equation (Rodell et al., 2011), following Eq. (3):

$$\upsilon_{ET} = \frac{\sqrt{\sigma_P^2 + \sigma_R^2 + \sigma_{\frac{dS}{dt}}^2}}{ET} \quad (3)$$

where $\sigma_P$, $\sigma_R$ and $\sigma_{\frac{dS}{dt}}$ respectively represent the absolute uncertainties in P, R and $\frac{dS}{dt}$. Errors in precipitation were estimated as the random error ($\sigma_{P\_random}$) plus the systematic error ($\sigma_{P\_bias}$), combined in quadrature. Random errors were calculated following Eq. (4), from Huffman (1997):

$$\sigma_{P\_random} = \bar{r}\left[\frac{H-p}{pN}\right]^{\frac{1}{2}} \quad (4)$$

where $\bar{r}$ is the climatological mean precipitation over the basin, $H$ is a constant (1.5), $p$ is the frequency of non-zero rainfall and $N$ is the number of independent precipitation samples (defined as the number of Amazon pixels with finite P measurements in each month). For $\sigma_{P\_bias}$, we used the value of –3.6 % estimated for CHIRPS from a validation analysis based on data from 21 meteorological stations in northeast Brazil (Table 4 in Paredes-Trejo et al., 2017). $\sigma_R$ was estimated as 5% of monthly river flow (Dingman, 2015). Uncertainty in groundwater storage was quantified by combining GRACE measurement errors and

leakage errors (residual errors after filtering and rescaling) in quadrature. For these, we used Amazon-specific values from the literature (6.1 and 0.9 mm for measurement and leakage errors, respectively) that had been calculated after CRI filtering and CLM scaling factors had been applied (Table 1 in Wiese et al., 2016). Finally, since $\frac{dS}{dt}$ values were calculated using data from two consecutive months, groundwater error values were multiplied by $\sqrt{2}$ to obtain $\sigma_{\frac{dS}{dt}}$ (e.g. Maeda et al., 2017). We calculated a mean $\upsilon_{ET}$ value of 16.1 % (standard deviation=9.2 %) for Amazon catchment-balance ET (Fig. S2). At the

monthly timescale, the $\frac{dS}{dt}$ and precipitation terms were found to be the dominant sources of uncertainty ($\sigma_{\frac{dS}{dt}}$=8.7 mm, $\sigma_P$=6.8 mm), followed by runoff ($\sigma_R$=4.9 mm, Table S2). Seasonal and interannual time series of precipitation, runoff, $\frac{dS}{dt}$ and ET, and their associated errors, are shown in Figures S3 and S4. Due to small interannual variation in $\frac{dS}{dt}$ (Fig. S4), climatological estimates of ET calculated with and without water storage estimates were similar (Figs. 1 & 2). Data from August 2017 to June 2018 were removed due to anomalously low and possibly unreliable $\frac{dS}{dt}$ data over this period (Fig. S4c). We tested the

sensitivity of our interannual trend analysis to the removal of these data points and found it had no statistically significant impact on the reported results.

## 2.2 Flux tower ET

To provide a ground-truth perspective, we used the 1999–2006 quality-controlled (QAQC), monthly flux tower ET observations from six flux towers in the LBA BrasilFlux database (Restrepo-Coupe et al., 2013, Saleska et al., 2013). These
data have been processed to remove unreliable or low-quality measurements, and can be downloaded from the LBA web page: https://daac.ornl.gov/LBA/guides/CD32_Brazil_Flux_Network.html. We selected towers situated over land-cover types that were representative of the surrounding area, including towers in forest, savanna and floodplain sites, but excluding towers in pasture sites where the dominant regional land cover was forest (see Table S3). The site locations are shown in Figure 1. We calculated ET in units of mm month$^{-1}$ (kg m$^{-2}$ month$^{-1}$) using Eq. (5):

$$ET = (LE/\lambda) \qquad (5)$$

where $LE$ is the monthly mean tower measurement of latent heat flux (W m$^{-2}$ = J s$^{-1}$ m$^{-2}$), scaled to J month$^{-1}$ m$^{-2}$, and $\lambda$ is the latent heat of vaporization at 20°C (2.453 x 10$^6$ J kg$^{-1}$).

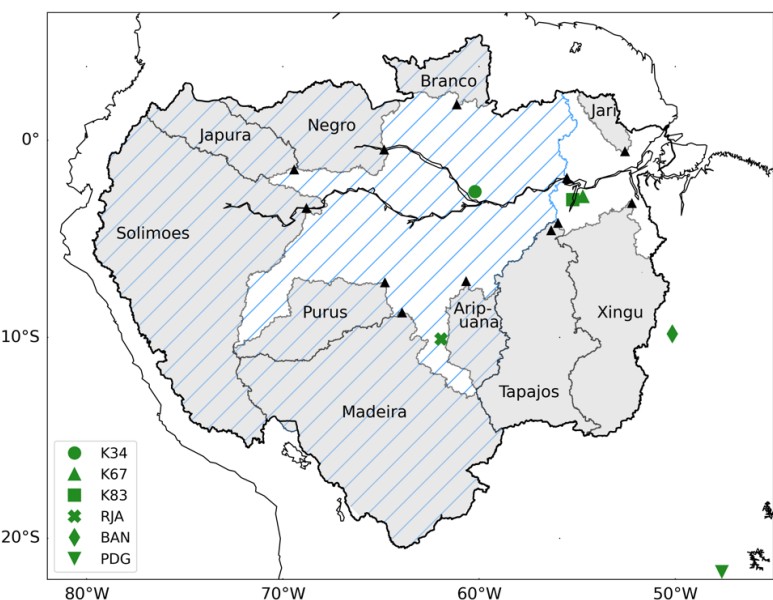

**Figure 1 – Locations of river catchments and in-situ data.** Map showing the locations of the Amazon sub-basins (grey polygons) and the respective river-gauge stations (black triangles) used to estimate catchment-balance ET. Note that two stations in the Tapajos basin were used (see Methods). Blue hatching indicates the area drained by the Óbidos measurement station, which is used to represent 'whole' Amazon ET. The locations of the LBA flux towers used in the study are also shown (green markers, see Table S3 for site information). The markers for K67 and K83 have been offset by 0.25° in longitude and
latitude, respectively, to improve visibility.

In addition to the QAQC LBA data, we used a unique 19-year record (1999–2017) from the K34 flux-tower site (2.6°S, 60.2°W) near Manaus, Brazil. Unlike the other tower sites, where data were only available for a few years (Table S3), this extended record could be used to derive a robust seasonal cycle in ET. Half-hourly data were averaged and scaled to obtain monthly means. To test the sensitivity of our results to missing data, we applied thresholds for the minimum number of hours or days required to calculate a mean value each month. Seasonal results were found to be relatively insensitive to minimum data requirement thresholds, thus we decided to include all monthly ET measurements in our analysis (Table S4).

## 2. 3 Satellite and reanalysis ET

Three global, satellite-derived ET products and one reanalysis dataset were included in this study. The Moderate Resolution Imaging Spectroradiometer (MODIS) MOD16A2 Version 6 ET product (Mu et al., 2011, Mu et al., 2013) was downloaded at 500-m resolution from the NASA Earth Data web page (https://earthdata.nasa.gov) for the period 2001–2019. The MODIS ET algorithm is based on the Penman-Monteith equation (Monteith, 1965), which uses temperature, wind speed, relative humidity and radiation data to approximate net ET, but modified by scaling canopy conductance by LAI. The sinusoidal 500-m MODIS tiles were merged and reprojected to a regular latitude-longitude grid (WGS84), using the Geospatial Data Abstraction software Library (GDAL/OGR Contributors, 2020), and resampling through weighted averaging. We also obtained ET estimates from the 8-km Process-based Land Surface ET/Heat Fluxes algorithm (P-LSH) product provided by the Numerical Terradynamic Simulation Group at the University of Montana (Zhang et al., 2010, Zhang et al., 2015a) for the period 1982–2013. This ET product is also based on the Penman-Monteith equation but uses an algorithm that incorporates remote-sensing NDVI data to estimate canopy conductance. Additionally, ET were retrieved from the satellite-constrained Global Land Evaporation Amsterdam Model (GLEAM) v3.3b dataset (Miralles et al., 2011, Martens et al., 2017) at 0.25° x 0.25°. This version of GLEAM is based on satellite data only, and available for the period 2003–2018. GLEAM is based on the Priestley-Taylor framework (Priestley and Taylor, 1972), which uses temperature and radiation to estimate potential ET (PET), together with a hydrological model to convert PET to actual ET. Finally, 0.25° x 0.25° ET data were retrieved for the period 2001–2019 from the European Centre for Medium-Range Weather Forecasts ERA5 reanalysis, which incorporates observations into a model to provide a numerical description of historical climate (Hersbach et al., 2018). Since ET is not among the many observations that are assimilated in the reanalysis, ERA5 ET is independent from the other ET datasets analysed in this study. To permit a meaningful comparison between datasets, all satellite and reanalysis ET products were re-gridded to 0.25°, analysed at the monthly timescale, and averaged over the common time period of 2003–2013 for those analyses based on temporal means. A summary of the equations and datasets used to derive the satellite ET products is presented in Table 1.

## 2.4 ET from coupled climate models

We obtained historical simulations of ET from models participating in CMIP5 and CMIP6 for the periods 1994–2004 and 2001–2014, respectively. We selected models that also provided precipitation, surface shortwave radiation and LAI output, in

order to investigate model processes controlling ET. In total, we used data from 13 CMIP5 models, and 10 CMIP6 models (Tables S5 and S6). Output was downloaded at monthly resolution from the Centre for Environmental Data Analysis archives (http://archive.ceda.ac.uk), accessed via the JASMIN supercomputer. Where available, multiple realisations were used to derive an ensemble mean for each model, else a single run was used. For basin-scale ET estimates, annual and seasonal climatological means were calculated for each model separately, using native-resolution data (see Tables S5 & S6), and then subsequently averaged across models. For CMIP5, climatologies were computed using data from 1994–2004 (most recent 11 years of available data), while CMIP6 climatologies were estimated using the same period as for observations (i.e. 2003–2013). To visualise spatial variation in ET over the Amazon and make comparisons with site-level ET measurements, multi-model ensemble means were also computed for CMIP5 and CMIP6. To do this, we re-gridded ET from each model to the same 1° x 1° horizontal grid and then calculated the ensemble mean across all models. Although not all models simulate the level of detail provided by 1° x 1° (see Tables S5 & S6 for native resolutions), this resolution enabled us to extract data from each Amazon sub-basin with more accuracy than using a coarser grid.

**Table 1 – Details of the evapotranspiration (ET) datasets analysed in this study.**

| ET data | Product(s) | Core equation | Input datasets | References |
|---|---|---|---|---|
| Catchment-balance | Computed in this study | $ET = P–R$ <br> or <br> $ET = P–R– dS/dt$ | – CHIRPS P <br> – R from ANA <br> – GRACE S | Funk et al. (2015) <br> HidroWeb (2018) <br> Wiese et al. (2018) |
| Satellite | MODIS MOD16A2 v6 | Penman-Monteith (Monteith, 1965) | – MODIS land cover (MOD12Q1) <br> – MODIS FPAR/LAI (MOD15A2) <br> – MODIS albedo (MOD43C1) <br> – GMAO v 4.0.0 reanalysis meteorology data | Mu et al. (2007) <br> Mu et al. (2011) |
| | P-LSH | Penman-Monteith (Monteith, 1965) | – AVHRR GIMMS NDVI <br> – NCEP/NCAR reanalysis meteorology data <br> – NASA GEWEX radiation <br> – FLUXNET tower data to parameterise canopy conductance model | Zhang et al. (2010) <br> Zhang et al. (2015a) |
| | GLEAM v3.3b | Priestley-Taylor (Priestley and Taylor, 1972) | – CERES L3 SYN1DEG Ed4A radiation <br> – AIRS L3 RetStd v6.0 air temperature <br> – MSWEP v2.2 precipitation <br> – GLOBSNOW L3Av2 & NSIDC v01 snow water equivalent <br> – LPRM vegetation optical depth <br> – ESA-CCI 4.5 soil moisture <br> – MEaSUREs VCF5KYR_001 vegetation fractions | Martens et al. (2017) |
| Reanalysis | ERA5 | Global model | A full list of input datasets is provided here: <br> https://confluence.ecmwf.int/display/CKB/ERA5%3A+data+documentation#ERA5:datadocumentation-Observations | Hersbach et al. (2018) |
| Climate model | CMIP5 | Global model | 13 Earth System Models (Table S5) | Taylor et al. (2012) |
| | CMIP6 | Global model | 10 Earth System Models (Table S6) | Eyring et al. (2016) |

**2.5 Dataset inter-comparison**

We compared differences in ET magnitude, spatial variation, seasonality and trends over the past two decades, identifying where estimates were in good agreement and where inconsistencies occurred. For annual comparisons, we computed climatological means over the Amazon basin (area drained by Óbidos, Fig. 2) and its sub-catchments (Fig. 3), using an area-weighted averaging approach. We applied the two-sample Kolmogorov-Smirnov test (Hodges, 1958) to identify whether monthly Amazon ET values from 2003 to 2013 from satellite, reanalysis and climate models were drawn from the same distribution as the catchment-balance ET values. We examined how well each ET product was able to capture spatial variation in Amazon ET, through comparisons with catchment-balance ET estimates and flux-tower measurements, and from correlating basin-scale annual means with catchment-balance ET (Table S7). ET products were also evaluated at the seasonal timescale over the Amazon catchment and at the K34 flux tower site (Fig. 1). For comparisons between flux tower and gridded ET data, we selected data from the single grid cell containing the tower.

All data were analysed over the period 2003–2013, with the exception of CMIP5, which was analysed over 1994–2004. The Amazon hydrological cycle has intensified between these periods, with increases in basin-mean P (Gloor et al., 2013), therefore, we might expect CMIP5 ET to show some differences from other ET products. However, results from CMIP5 were largely consistent with results from CMIP6, showing that any differences caused by analysis time period were smaller than the differences between the models and other types of ET data. We acknowledge that the period for evaluating Amazon ET is relatively short, though we were constrained by our reliance on satellite data and the availability of climate model output.

We also analysed linear trends in Amazon basin ET, using data averaged across all months (annual), the wettest three months (January–March, JFM) and the driest three months (July–September, JAS) over the past two decades using ordinary least squares regression. Years with fewer than ten months of data were excluded from the annual time series (2017 and 2018), and years with any missing months in JFM or JAS were excluded from the wet and dry season time series (2017 in JAS only). Trends were analysed over the time period common to all datasets (2003–2013), and across all years with available data for each dataset.

**2.6 Investigating controls on Amazon ET**

To better understand differences between ET products, we analysed relationships with potential drivers of ET, including precipitation, surface radiation and LAI. Satellite-based ET estimates were compared with precipitation from CHIRPS, radiation from CLARA-A1 (Karlsson et al., 2013) and LAI from the quality-controlled MODIS MOD15A2H Collection 6 (C6) product provided by Boston University (Myneni et al., 2015, Yan et al., 2016b), each re-gridded to 0.25° x 0.25°. MODIS LAI has been shown to perform relatively well against ground-based LAI measurements ($R^2$=0.7–0.77), though uncertainty over the validity of high LAI values (>4 $m^2$ $m^{-2}$), such as occur over the Amazon, is larger due there being few ground

measurements and the satellite reflectance signal reaching saturation over dense canopies (Yan et al., 2016a). Furthermore, the satellite-based MODIS ET product incorporates MODIS LAI (Table 1), and thus these datasets are not fully independent from one another. CLARA-A1 radiation is independent from the ET datasets evaluated in this study and estimated to have an accuracy of $\leq 10$ W m$^{-2}$, though few validation measurements were available over South America, and none in the Amazon region (Karlsson et al., 2013). Thus, there is some uncertainty in the accuracy of these satellite products over the Amazon that must be considered when interpreting the results. Reanalysis and model ET were compared with reanalysis and model variables, respectively. For ERA5, we used the 'high vegetation' LAI field since the Amazon is predominantly covered with tropical forest, though repeating the analysis with 'low vegetation' LAI made little difference to the results. For the K34 tower site, ET was compared against precipitation and radiation data only. Half-hourly measurements of precipitation and incoming radiation from the tower site were averaged and scaled to monthly resolution, following the same procedures as applied to the ET data. Due to missing data in several years, climatological means and seasonal cycles for K34 were calculated using all data from 1999–2017.

We analysed controls on spatial variation in ET by comparing catchment-mean values against catchment means of precipitation, radiation and LAI. Since there were only eleven data points in this analysis (representing the Amazon and ten sub-catchments), we also analysed the response of ET to spatial variation in its potential drivers at the grid-cell level, following a similar approach to Ahlström et al. (2017). This enabled us to better understand non-linear relationships between ET and its controlling variables. Mean annual ET values from all Amazon grid cells were binned by annual precipitation, radiation and LAI using bin widths of 100 mm, 5 W m$^{-2}$ and 0.2 m$^2$ m$^{-2}$ respectively. Bins with fewer than five data points were excluded from the analysis. Finally, to distinguish between the controls on seasonal variation in ET from the controls in interannual variation in ET, we analysed relationships between ET and possible driving variables at the climatological monthly timescale and at the interannual timescale. While this approach was useful to understand the relative importance of controlling variables at different timescales, it reduced the number of data points in each analysis such that statistical power was correspondingly low. This meant that when we did not detect a statistically significant signal then it could either be because there was no signal to detect, or because the signal-to-noise ratio was too low. This should be taken into consideration when assessing the analysis of controls on Amazon ET reported here.

## 3 Results and Discussion

### 3.1 Comparing estimates of annual ET over the Amazon

Climatological annual Amazon ET estimates from water-balance approaches, satellite-based products, reanalysis and two coupled-model ensembles are presented in Figure 2. ET from catchment-balance was the lowest of all estimates (mean ± standard deviation = 1083±37 mm year$^{-1}$ for 2003–2013, Fig. 2, Table S7), which, given uncertainties, is indistinguishable from the value obtained from differencing precipitation and runoff (1102±53 mm year$^{-1}$). This confirms that the GRACE-

observed changes in groundwater storage are relatively small over decadal timescales. Our mean annual catchment-balance ET estimate for the Amazon was very similar to that from a previous catchment-balance study (1058 mm yr[-1]), calculated over the same drainage region (drained by Óbidos) but based on different precipitation data and an alternative GRACE solution (Swann and Koven, 2017), suggesting the approach is relatively robust. The area drained by Óbidos excludes the far eastern Amazon, which our spatial catchment-balance analysis revealed to have the highest annual ET across the basin, decreasing towards the west and south (Fig. 3a, b). This may explain why our catchment-balance annual Amazon ET value was towards the lower end of previous estimates (Marengo, 2006).

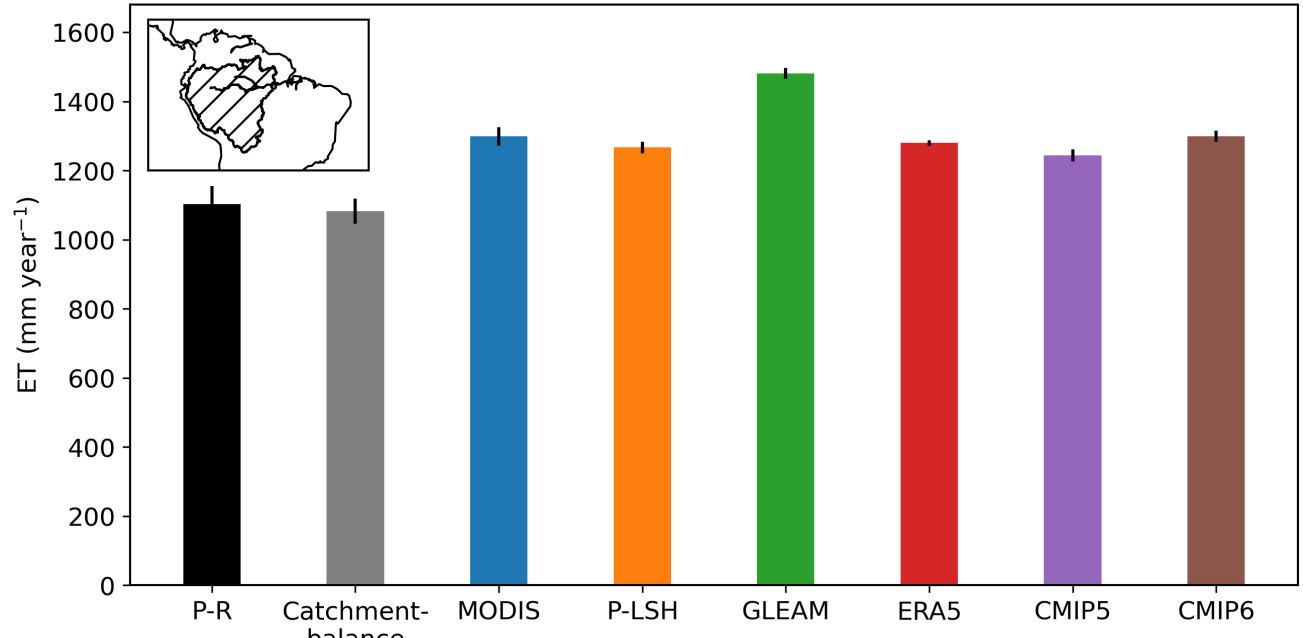

**Figure 2 – Comparison of annual Amazon evapotranspiration (ET) estimates.** Climatological mean Amazon ET estimated from water-balance approaches (precipitation minus runoff, P–R and catchment-balance accounting for change in groundwater storage, P–R–$\frac{dS}{dt}$), satellites (MODIS, P-LSH, GLEAM), ERA5 reanalysis, and climate models (CMIP5 and CMIP6). Data are from 2003 to 2013, with the exception of CMIP5, where the data are from 1994–2004. Error bars represent the interannual standard deviation for each dataset. For CMIP5 and CMIP6 the error bars represent the average standard deviation across all models. Data from satellites, reanalysis and models were averaged over the region shown in the inset map for a direct comparison with the water-balance approaches.

Annual Amazon ET from satellites, reanalysis and coupled models was 15–37 % higher than catchment-balance ET, with GLEAM showing the largest bias (Fig. 2). With the exception of GLEAM, mean annual ET values from satellites, reanalysis and coupled models were remarkably similar to one another (within 50 mm, or <4 %), with a mean bias of 18 % (relative to

catchment-balance ET). ET from all of the products and models analysed showed statistically different distributions from catchment-balance ET (Kolmogorov-Smirnov test, Fig. S5a), tending to show a narrower range and fewer low ET values (Fig. S5b). This substantial and consistent overestimation of annual Amazon ET across data products and coupled models highlights that even basic features of the Amazon hydrological cycle are still not well characterised.

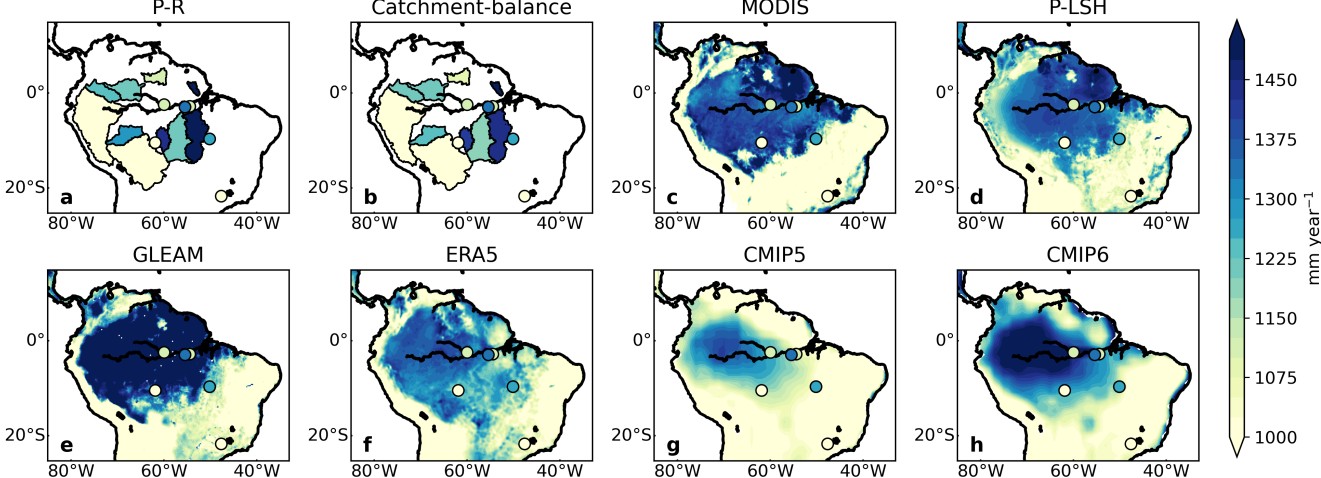

**Figure 3 – Spatial variation in Amazon evapotranspiration from different approaches.** Climatological mean annual ET from (a) differencing precipitation and runoff, (b) catchment-balance analysis accounting for change in groundwater storage, (c–e) satellite-based ET products, (f) ERA5 reanalysis, and (g, h) climate models. Coloured circles on each panel indicate ET measured at six flux tower sites. Where there were multiple tower sites in close proximity, circles were plotted with an offset of 0.5° to improve data visualisation. Data for panels a–f and h are from 2003–2013, data for panel g are from 1994–2004 and flux-tower data are from the periods shown in Table S3. Data in panels c–h are plotted as contour maps with contours at 25-mm intervals from 1000 to 1500 mm year$^{-1}$. GLEAM data are presented with an alternative scale in Figure S6.

MODIS and P-LSH captured a northeast to southwest gradient in ET across the basin that was evident in the water-balance approaches, showing highest ET over the Guiana shield in the north of the Amazon and decreasing southwest across the basin (Fig. 3c, d). Catchment-mean ET values from these two products were strongly correlated with ET from the catchment-balance approach across the eleven basins analysed in this study (r=0.84, p<0.01 and r=0.82, p<0.01 for MODIS and P-LSH, respectively), though spatial variability was weaker and interannual variability was also strongly underestimated (Fig. S7,
Tables 2 and S7). Flux tower ET measurements, although spatially limited, appear to show an east-west gradient in Amazon ET, with the highest annual values over forest and seasonally-flooded sites in the east of the basin (coloured circles in Fig. 3). However, the gradient in tower data should be interpreted with some caution, since variation in energy-balance closure between sites will affect the absolute ET values (da Rocha et al., 2009a, Fisher et al., 2009). Furthermore, two nearby towers in the northeast Amazon showed a clear difference in mean annual ET (K67 & K83), likely due to being located on different land

cover types (primary forest and selectively logged forest, respectively, Table S5). ET from GLEAM, which exceeded 1400 mm year$^{-1}$ over much of the Amazon, showed a north-south ET gradient (Fig. 3e, see Fig S6 for alternative scale), and a positive, though not statistically significant, correlation with catchment-balance estimates (r=0.51, p=0.11, Fig. S7, Table S7). Previous studies based on flux-tower measurements (Fisher et al., 2009), water-budget analysis (Zeng et al., 2012, Maeda et al., 2017, Sun et al., 2019), and a combination of satellites and flux towers (Paca et al., 2019) showed similar north/northeast to south/southwest gradients in ET across the Amazon, in line with the catchment-balance results presented in Figure 3.

ET from ERA5, CMIP5 and CMIP6 bore no relation to catchment-balance ET, simulating the highest ET values in the northwest of the basin and decreasing to the east (Fig. 3e–g). The CMIP models do not incorporate any observations, and therefore might not be expected to perform as well the other products analysed in this study. However, an analysis of Amazon precipitation in 11 CMIP5 models found that most were able to capture spatial patterns relatively well, including shifting distributions through the course of the seasonal cycle (Yin et al., 2013). The poor representation of spatial variation in Amazon ET in reanalysis and coupled models shown in Figure 3 demonstrates a need for improvement of this key hydrological variable in these products.

To understand the drivers of spatial variation in Amazon ET, we compared catchment-scale estimates against catchment-means of precipitation, surface radiation and LAI (Fig. 4). Since there were only eleven data points in the analysis (representing the Amazon and ten sub-catchments), statistical power was relatively low. However, we found spatial variation in catchment-balance ET showed some indication of an influence from radiation (r=0.38, p=0.25, Fig. 4h), but not precipitation (r=0.14, p=0.68, Fig. 4a) or LAI (r=0.06, p=0.87, Fig. 4o). This result tentatively suggests that spatial variation in radiation explains more of the spatial variability in ET across Amazon sub-catchments than other variables. None of the ET products and models analysed captured positive relationships between catchment-mean ET and radiation. ET from ERA5 and the CMIP ensembles instead showed negative associations with radiation (Fig. 4l–n), and, along with GLEAM ET, positive relationships with precipitation (Fig. 4d–g), indicative of water availability influencing spatial variation in ET (Fig. 4d–g). These results confirm that the reanalysis and climate models analysed here struggled to capture spatial patterns in Amazon ET due to misrepresentation of the controlling drivers, specifically the relative importance of precipitation and net radiation. ET from ERA5 and the models also showed positive correlations between LAI and ET (Fig. 4s–u), not seen in the satellite observations. However, it should be noted that satellite LAI was generally slightly lower and showed less spatial variability than other LAI datasets over the Amazon (Fig. S8i–l), likely due to the satellite sensor being insensitive to variation in LAI over areas of dense tropical forest (Myneni et al., 2002, Yan et al., 2016a). This could hamper our ability to accurately assess the extent to which LAI influences spatial variation in ET.

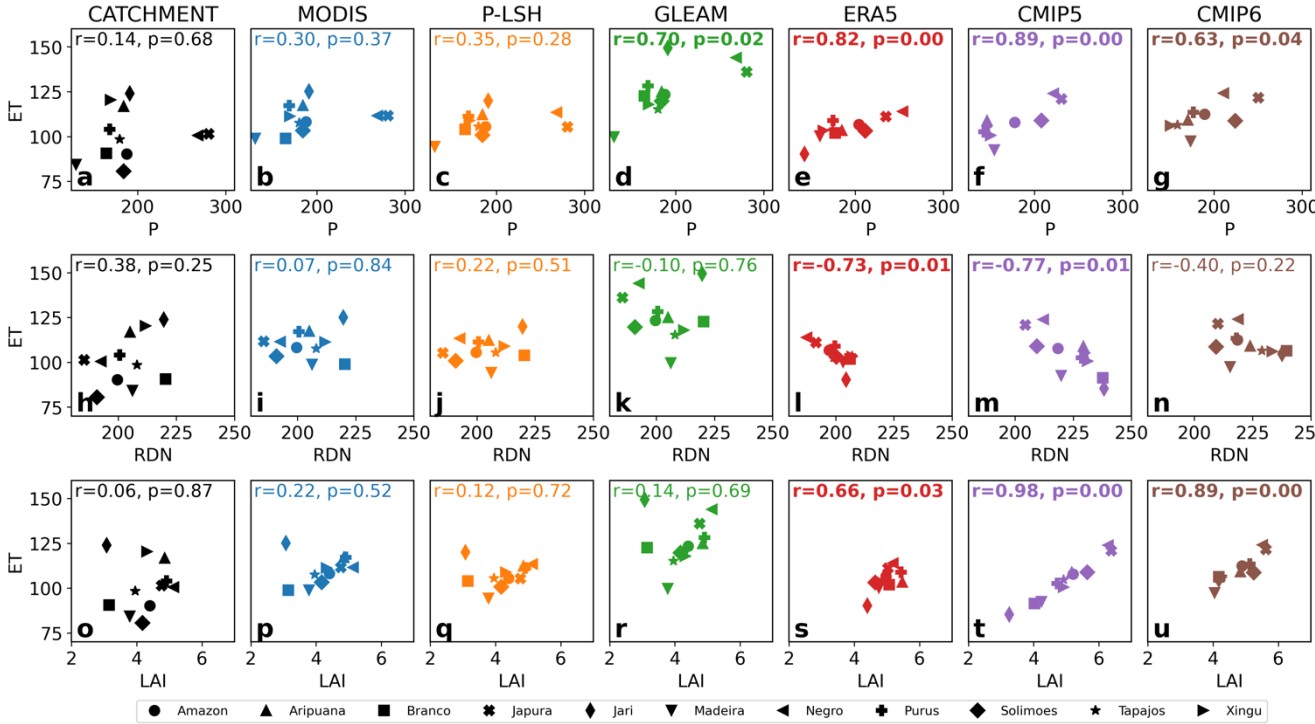

**Figure 4 – Controls on spatial variation in Amazon evapotranspiration.** Annual mean ET (in mm month[-1]) for the Amazon and ten sub-catchments (Fig. 1) from catchment-balance, satellites (MODIS, P-LSH, GLEAM), ERA5 reanalysis, and climate models (CMIP5 and CMIP6), plotted against (a–g) precipitation (P, mm month[-1]); (h–n) surface shortwave radiation (RDN, W m[-2]); and (o–u) leaf area index (LAI, m[2] m[-2]). Satellite ET data are plotted against P from CHIRPS, RDN from CLARA-A1

and LAI from MODIS; ERA5 and climate model ET are plotted against ERA5 and model P, RDN and LAI, respectively. Data are from 2003 to 2013, with the exception of CMIP5, where the data are from 1994–2004. Note that the axes do not start at zero.

For further insights on the validity of Amazon ET products and the factors controlling ET, we evaluated ET responses to spatial

variation in precipitation, radiation and LAI at the grid-cell level (Fig. 5). Differences between ET products were most apparent in their responses to annual precipitation (Fig. 5a). Above 2000 mm yr[-1] datasets followed three patterns of behaviour: GLEAM ET continued to increase to approximately 1600 mm yr[-1], ET from MODIS, P-LSH and ERA5 remained relatively stable at around 1300 mm yr[-1], and CMIP5 and CMIP6 showed slight reductions in ET with further increases in precipitation. The precipitation threshold of 2000 mm yr[-1] has previously been suggested as the level above which tropical forests are able to

sustain photosynthesis during the dry season (Guan et al., 2015), and as the breakpoint between productivity in the Amazon being water (< 2000 mm yr[-1]) or radiation (> 2000 mm yr[-1]) limited (Ahlström et al., 2017). Indeed, below 2000 mm yr[-1] ET increased with increasing precipitation for all satellite, reanalysis and model datasets (lines in Fig. 5a), indicating a water-

limitation on ET. The two catchments in the northwest Amazon where P exceeds 3000 mm yr[-1], Japura and Negro, were most closely aligned with the products that showed ET levelling off when precipitation exceeded 2000 mm yr[-1] (i.e. MODIS, P-LSH and ERA5), suggesting these products represent the ET response to rainfall in very wet areas relatively well. For MODIS and P-LSH, this finding provides additional support that spatial patterns in Amazon ET correspond well with spatial variation in its controlling variables. In contrast, although ERA5 generally captured the correct ET response to precipitation (Fig. 5a), there are spatial differences between satellite and ERA5 precipitation datasets in Amazon regions with rainfall above 2000 mm yr[-1] (Fig. S8a–d). This explains why relationships between ERA5 precipitation and ET differed at the catchment (Fig. 4e) and the grid-cell (Fig. 5a) scales. In the GLEAM model, the 'stress factor' that is used to scale PET takes precipitation as an input variable to the soil module (Table 1), which in turn controls the amount of water available for ET (Martens et al., 2017). Our results indicate that the GLEAM model overestimates the dependence of ET on soil moisture in regions with high annual rainfall, highlighting a possible target for improvements to the GLEAM algorithm.

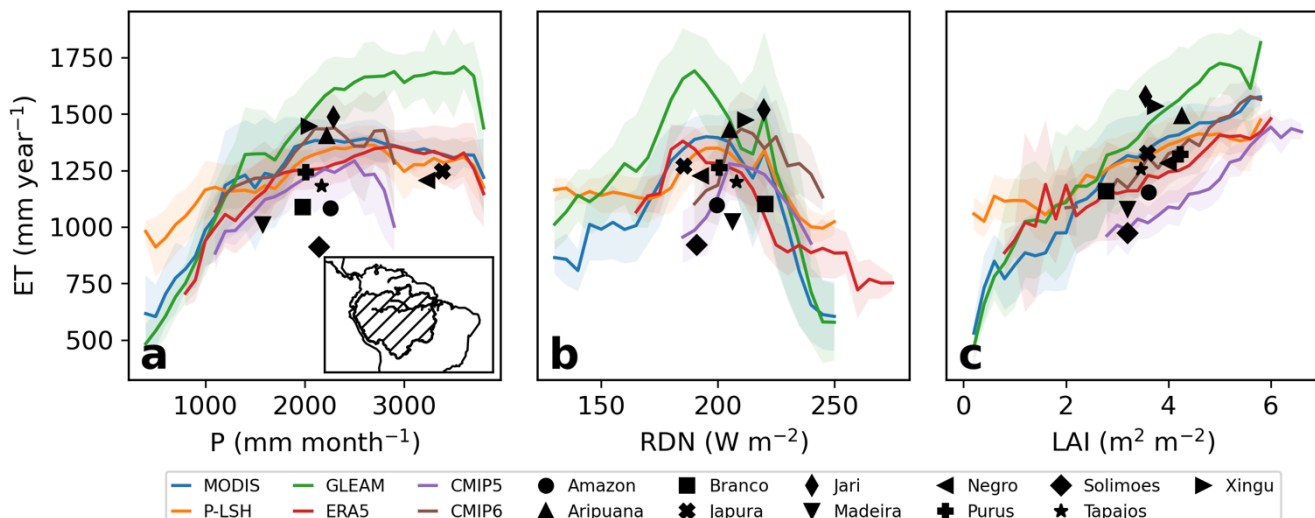

**Figure 5 – ET response to spatial variation in controls.** ET data from satellites (MODIS, P-LSH, GLEAM), ERA5 reanalysis, climate models (CMIP5 and CMIP6), and catchment-balance (black markers) are plotted against annual (a) precipitation (P); (b) surface shortwave radiation (RDN); and (c) leaf area index (LAI). Shading represents the standard deviation of the mean. Satellite ET data are plotted against P from CHIRPS, RDN from CLARA-A1 and LAI from MODIS; ERA5 and climate model ET are plotted against ERA5 and model P, RDN and LAI, respectively. Data were extracted from the Amazon region indicated in the inset map in panel a. The locations of the catchments and tower sites are indicated in Figure 1. Data are from 2003 to 2013, with the exception of CMIP5, where the data are from 1994–2004. Note that the axes do not start at zero.

Differences between ET products in their relationships with other variables were more subtle. ET dependence on radiation was broadly similar among datasets, showing a peak at approximately 200 W m$^{-2}$ (Fig. 5b). This is consistent with low and high levels of radiation tending to correspond with high and low levels of precipitation, respectively (Fig. S8a–h), and ET peaking at an optimum between the two. LAI-ET relationships were also fairly consistent, with ET increasing relatively linearly with

increasing LAI (Fig. 5c). GLEAM generally tended to overestimate ET relative to LAI, while CMIP5 underpredicted ET for a given LAI value, in comparison with ET from other products and catchment-balance estimates. In general, radiation over the Amazon was substantially higher in the models compared to satellite and reanalysis datasets (Fig. S8a–h), and satellite-derived LAI values were uniformly lower than other estimates (Fig. S8a–h), likely due to signal saturation (Myneni et al., 2007).

**Table 2 – Summary of comparative statistics.** Datasets listed in the table were correlated with catchment-balance ET estimates (spatial, seasonal and interannual), and interannual standard deviations ($\sigma$) were calculated over the period 2003–2013 using data standardised by the climatological mean over that period (units of mm month$^{-1}$ year$^{-1}$). Statistically significant ($p<0.05$) relationships are shown in bold. Note that CMIP data were not correlated at the interannual scale because model years would not be expected to align with real-world years.

| ET dataset | Spatial | Seasonal | Interannual variability, correlations with catchment balance and trends over 2003–2013 (mm month$^{-1}$ year$^{-1}$) | | | | | | | | |
| | Climatological catchment means | Amazon | Annual | | | Wet (JFM) | | | Dry (JAS) | | |
| | | | $\sigma$ | r | Slope | $\sigma$ | r | Slope | $\sigma$ | r | Slope |
| Catchment-balance | - | - | 2.90 | - | -0.09 | 8.89 | - | -0.74 | 8.92 | - | 0.15 |
| MODIS | **0.84** | **0.63** | 2.28 | -0.24 | **-0.58** | 5.56 | 0.19 | **-1.30** | 5.01 | -0.34 | 0.23 |
| P-LSH | **0.82** | **0.67** | 1.37 | -0.11 | **0.41** | 1.83 | 0.00 | -0.05 | 4.09 | -0.09 | **0.88** |
| GLEAM | 0.51 | -0.18 | 1.36 | -0.42 | 0.09 | 1.82 | -0.44 | 0.07 | 4.90 | -0.36 | 0.41 |
| ERA5 | -0.28 | **0.61** | 0.65 | 0.13 | 0.05 | 1.91 | 0.01 | -0.30 | 1.21 | **-0.51** | -0.11 |
| CMIP5 | -0.06 | -0.11 | - | - | - | - | - | - | - | - | - |
| CMIP6 | -0.14 | 0.05 | 0.48 | - | 0.02 | 0.37 | - | 0.05 | 1.48 | - | 0.07 |

### 3.2 Seasonal variation in Amazon ET

The mean seasonal cycle in Amazon ET was estimated from catchment-balance analysis, satellite, reanalysis and model ET datasets for the whole Amazon basin (Fig. 6). Amazon catchment-balance ET showed a strong seasonal cycle (standard deviation, $\sigma=22$ mm month$^{-1}$), with annual minima during April–June and maxima in August–October (Fig. 6). ET at the K34

tower site, located in the central Amazon, showed a similar seasonal pattern to that over the wider basin (Fig. S9), though intra-annual variation was weaker ($\sigma=14$ mm month$^{-1}$). Furthermore, we observed strong, positive correlations between ET and radiation for the Amazon basin and K34 tower site (r=0.93, $p<0.001$ and r=0.68, $p<0.05$, respectively, Figs. 7h, S10), and

between ET and LAI for the basin (r=0.63, p<0.05, Figs. 7o, S11). These results agree with findings from da Rocha et al. (2009a), who made a detailed comparison of seasonal ET at seven flux tower sites in Brazil. They showed ET increased during the dry season at the four wet tropical forest sites (including K34), contrasting with three transition-forest and savanna sites where ET followed seasonal soil moisture availability. The seasonal cycle in ET shown in Figure 6 is consistent with studies reporting an increase in leaf flush driving Amazon greening in the dry season (Lopes et al., 2016, Saleska et al., 2016). Studies based on catchment-balance analysis (Swann and Koven, 2017), and satellite observations of vegetation photosynthetic properties (Guan et al., 2015) also showed that ET and forest productivity peak during the drier part of the year over the majority of the Amazon. Finally, our results are in agreement with those from Fisher et al. (2009), who identified radiation and NDVI as the primary and secondary controls on ET across the tropics based on analysis of flux tower measurements.

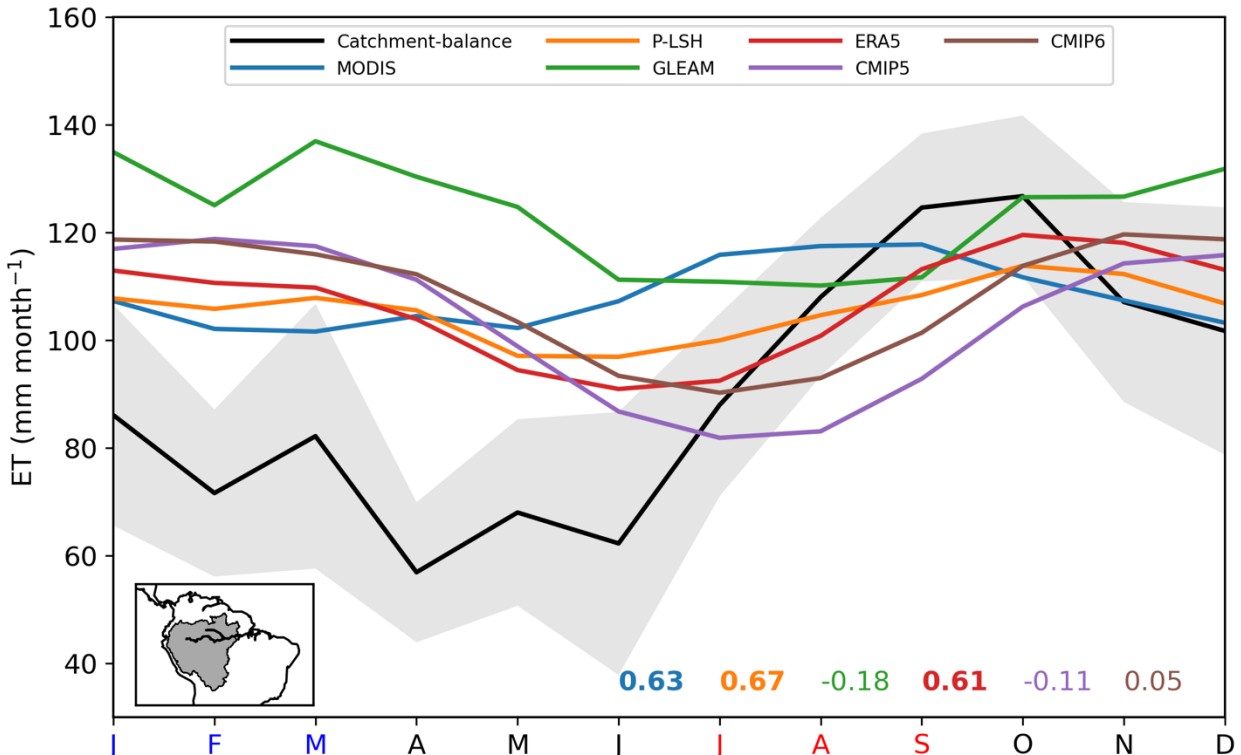

**Figure 6 – Climatological seasonal cycles in evapotranspiration over the Amazon.** Mean seasonal cycle in ET from catchment balance, satellites (MODIS, P-LSH, GLEAM), ERA5 reanalysis and climate models (CMIP5 and CMIP6) over the Amazon region drained by Óbidos (region indicated in the inset map). Shading represents the monthly standard deviation of the mean. Correlations with catchment-balance ET are shown, with bold numbers indicating statistical significance (p<0.05). Data are from 2003 to 2013, with the exception of CMIP5, where the data are from 1994–2004. On the x-axis, the three wettest months are indicated in blue and three driest months are indicated in red. Note that the y axis does not start at zero.

Monthly ET cycles from MODIS, P-LSH and ERA5 correlated with Amazon catchment ET (r=0.61–0.67, p<0.05, Table 2, Fig. 6), and captured positive relationships with surface radiation (r=0.66–0.78, p<0.05, Fig. 7). However, despite representing the direction of seasonal fluctuations relatively well, these datasets underestimated the seasonal variability by 39–77 %, relative to catchment-balance ET (Fig. 6). Biases from catchment-balance ET were generally strongly positive from January to June and weakly negative in September and October. At K34, MODIS and ERA5 overestimated the seasonal ET range by 61 and 28 %, respectively, while P-LSH underestimated the range by 26% (Fig. S9). With such poor representation of the magnitude of seasonal variability, and inconsistencies in the direction of amplitude biases, ET from these satellite and reanalysis datasets may be of limited use for assessing long-term changes in the seasonality of the Amazon hydrological cycle (Gloor et al., 2013), or for evaluating seasonal ET representation in coupled climate models.

ET from GLEAM, CMIP5 and CMIP6 neither correlated with seasonal catchment-balance Amazon ET, nor captured the correct seasonal amplitude (Figs. 6, 7, Table 2). Instead, ET from these datasets followed the same seasonal cycle as precipitation, peaking during the wettest part of the year. A previous study comparing Amazon ET estimates derived using different methods also observed that climate model and reanalysis ET tended to follow the precipitation seasonal cycle, with annual ET minima in the dry season (Werth and Avissar, 2004). The authors suggested this was due to a strong vegetation control on modelled ET due to down-regulation of stomatal conductance in the dry season, concluding such a control to be as credible as a radiation control on Amazon ET. However, a subsequent study queried this assertion, citing evidence from flux towers as proof that vegetation controls on Amazon ET were secondary to environmental controls (Costa et al., 2004). Over the Congo, where ET follows the same seasonal cycle at precipitation, CMIP5 models were shown to capture the seasonality of ET but overestimated the magnitude of the flux, particularly during the two wet seasons (Crowhurst et al., 2020). The results presented in Figure 6 indicate a disconnect between our mechanistic understanding of the controls on seasonal Amazon ET based on catchment-balance analysis, and the algorithms used to predict ET in GLEAM and the CMIP models.

Northern and southern Amazon sub-basins were analysed separately, due to differences in the timing of the seasonal precipitation cycle above and below the equator. Uncertainties in monthly ET estimates were higher over these areas than over the whole Amazon, though it was still possible to detect differences between catchment-balance ET and other datasets (Fig. S12). The seasonal cycle in catchment-balance ET was weaker in the north than the south ($\sigma$=16 vs. 26 mm month$^{-1}$, respectively), following the pattern of precipitation seasonality ($\sigma$=69 vs. 115 mm month$^{-1}$ in northern and southern basins, respectively). In general, satellite, reanalysis and climate model ET related fairly well to seasonal catchment-balance ET in the northern Amazon (Fig. S12b), but showed much weaker relationships in the southern Amazon (Fig. S12c). The CMIP5 and CMIP6 models, which were unable to capture seasonal ET variation over the whole Amazon or southern Amazon, replicated month-to-month variation in ET over the northern Amazon well, although both model groups underestimated seasonal variability (Fig. 6, S12). MODIS, which captured seasonal ET over the whole Amazon (Fig. 6), performed especially poorly in the south, showing a negative relationship with catchment-balance ET (r=−0.57 p=0.06, Fig. S12b). These results suggest

the ability of ET products to capture seasonal ET varies regionally, and a product that performs well over one region may not be reliable elsewhere. Finally, we note that relative uncertainties in ET estimated using the catchment balance approach increase at smaller spatial scales, precluding a more in-depth assessment of seasonal ET over individual sub-basins.

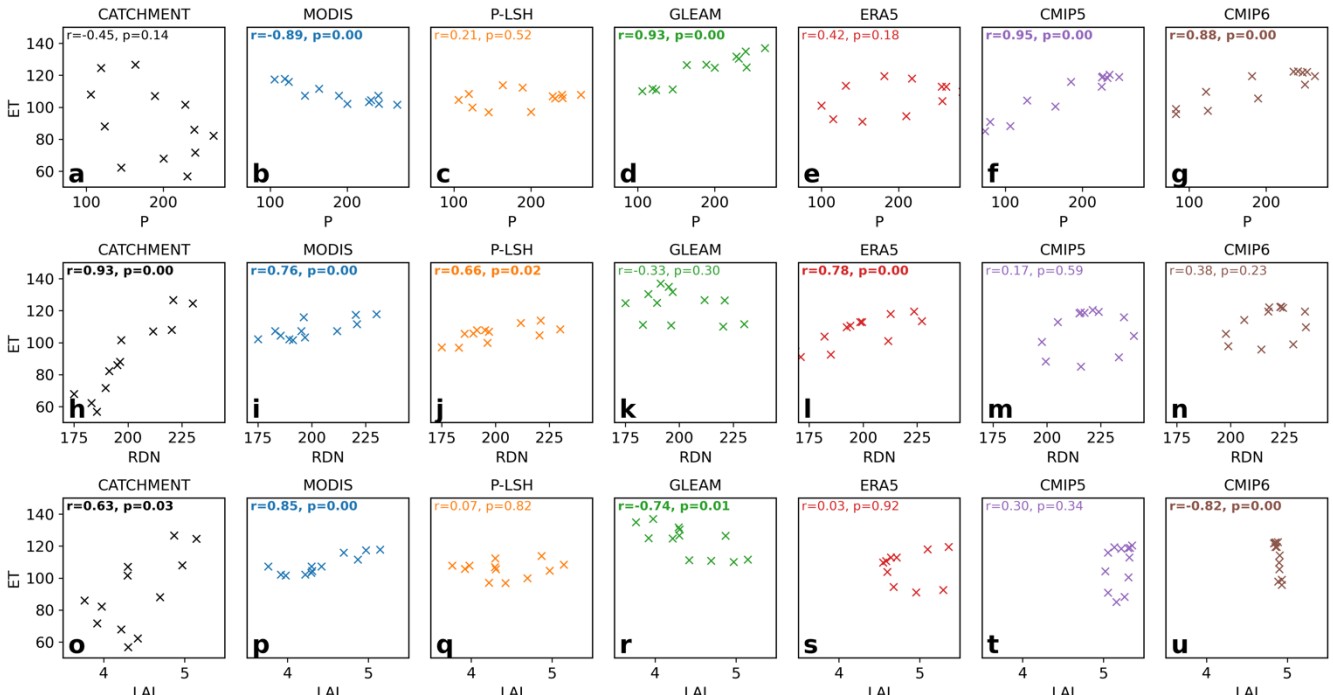

**Figure 7 – Controls on seasonal variation in Amazon evapotranspiration.** Monthly ET (in units of mm month$^{-1}$) for the Amazon region drained by Óbidos (see Fig. 1) from catchment-balance, satellites (MODIS, P-LSH, GLEAM), ERA5 reanalysis, and climate models (CMIP5 and CMIP6) plotted against (a–g) precipitation (P, mm month$^{-1}$); (h–n) surface shortwave radiation (RDN, W m$^{-2}$); and (o–u) leaf area index (LAI, m$^2$ m$^{-2}$). Satellite ET data are plotted against P from CHIRPS, RDN from CLARA-A1 and LAI from MODIS; ERA5 and climate model ET are plotted against ERA5 and model 560  P, RDN and LAI, respectively. Data are from 2003 to 2013, with the exception of CMIP5, where the data are from 1994–2004. Note that the axes do not start at zero.

### 3.3 Interannual variation and trend analysis

Interannual time series of Amazon ET from 2001 to 2019 for the whole year, the three wettest months (JFM, see Fig. S10) and the three driest months (JAS), are shown in Figure 8. From 2003 to 2013, interannual variability (σ) in catchment-balance ET 565  was 2.9 mm month$^{-1}$, or 3.2 % of the climatological mean. This value is comparable to the interannual variation in precipitation over the same period (σ=3.6 %), half the variation in runoff (σ=7.0 %) and represents around a tenth of the seasonal variation in Amazon ET (Fig. 6). With only a relatively short time series, controls on interannual variability were hard to detect, though

radiation appeared to play a role (Fig. S13). Interannual variation was underestimated in ET from satellites, reanalysis and climate models by up to a factor of six relative to catchment-balance (Fig. 8a, Table 2). In JFM and JAS, ET variation was

higher than at the annual scale (catchment-balance $\sigma$ = 8.89 and 8.91 mm month$^{-1}$, respectively) and similarly underestimated by other datasets (Fig. 8b,c, Table 2). Relationships between interannual catchment-balance ET and ET from satellites or reanalysis were generally poor (Table 2), and an especially high JFM catchment-balance ET recorded in 2016, coinciding with a severe El Niño event (Koren et al., 2018), was not captured by other ET products (Fig. 8b). ERA5 and CMIP6 showed the least interannual variation, indicating poor model representation of the factors influencing inter-year changes in ET.

**Table 3 – Interannual trends in Amazon ET.** Linear trends in annual, January–March (JFM) and July–August (JAS) ET were calculated over the time period common to all datasets (2003–2013) and for all years with available data in the past two decades (units of mm month-1 year-1). Statistically significant (p<0.05) trends are shown in bold.

| ET dataset | Time period | Annual | | JFM | | JAS | |
|---|---|---|---|---|---|---|---|
| | | Slope | p-value | Slope | p-value | Slope | p-value |
| **Catchment-balance** | 2003–2013 | -0.09 | 0.77 | -0.74 | 0.43 | 0.15 | 0.88 |
| | 2003–2019 | -0.10 | 0.60 | 0.27 | 0.66 | -0.51 | 0.28 |
| **MODIS** | 2003–2013 | **-0.58** | **0.00** | **-1.30** | **0.01** | 0.23 | 0.68 |
| | 2001–2019 | -0.21 | 0.07 | **-0.52** | **0.04** | 0.21 | 0.38 |
| **P-LSH** | 2003–2013 | **0.41** | **0.00** | -0.05 | 0.80 | **0.88** | **0.02** |
| | 2001–2013 | **0.32** | **0.00** | -0.01 | 0.94 | **0.63** | **0.03** |
| **GLEAM** | 2003–2013 | 0.09 | 0.56 | 0.07 | 0.74 | 0.41 | 0.43 |
| | 2003–2017 | -0.32 | 0.09 | **-0.50** | **0.02** | 0.12 | 0.73 |
| **ERA5** | 2003–2013 | -0.05 | 0.47 | -0.30 | 0.13 | -0.11 | 0.40 |
| | 2001–2019 | 0.01 | 0.86 | **-0.27** | **0.03** | 0.03 | 0.69 |
| **CMIP6** | 2003–2013 | 0.02 | 0.77 | 0.05 | 0.15 | 0.07 | 0.66 |
| | 2001–2014 | 0.00 | 0.96 | 0.03 | 0.42 | -0.02 | 0.88 |

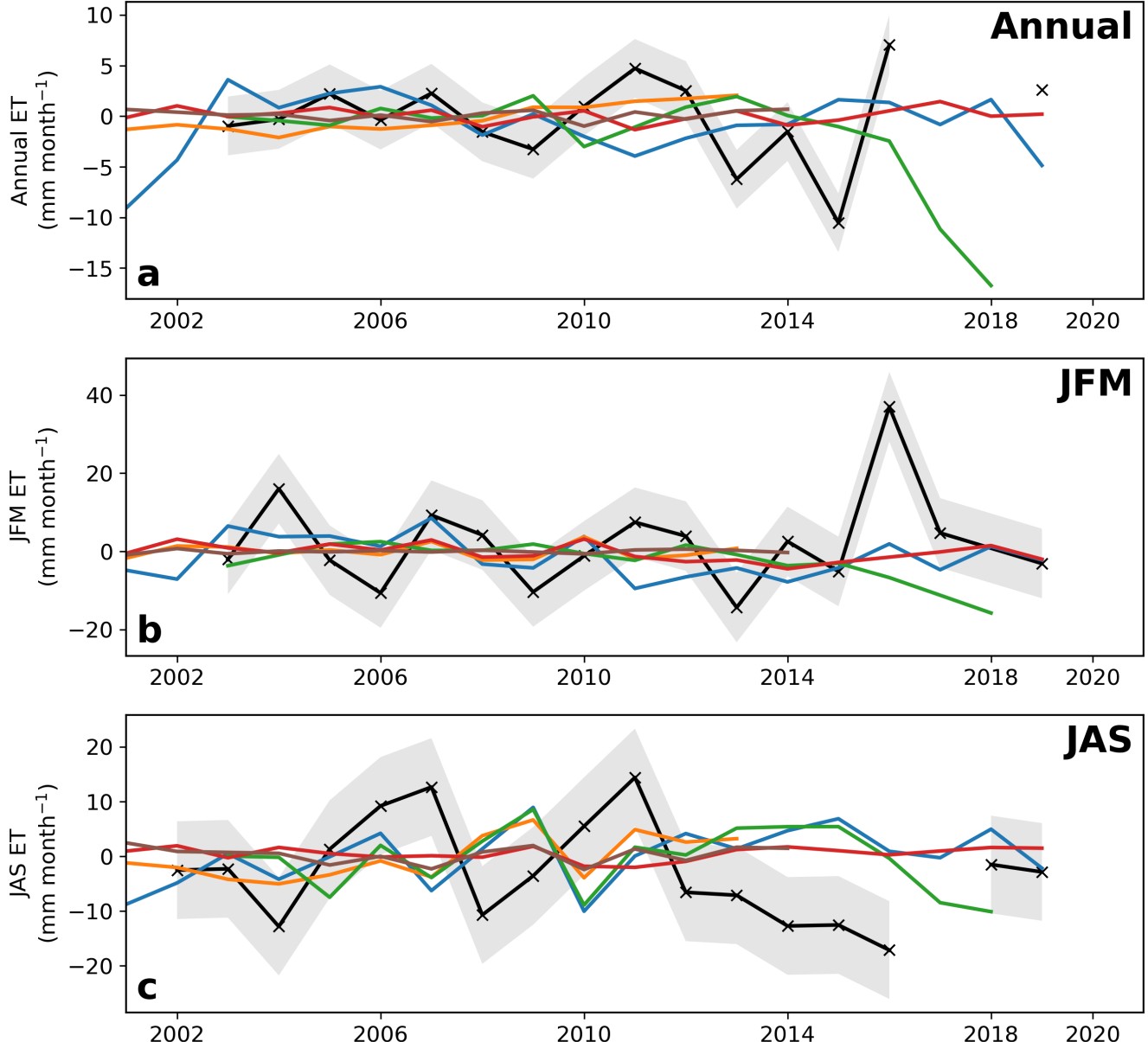

**Figure 8 – Interannual variation in evapotranspiration from 2001 to 2019.** Time series in ET over the Amazon from catchment balance (black, region drained by Óbidos, Fig. 1), satellites (MODIS, P-LSH, GLEAM), ERA5 reanalysis and CMIP6 models, for (a) the whole year, (b) January–March (JFM) and (c) July–September (JAS), normalised by the 2003–2013 climatological mean. Interannual trends are listed in Table 3. Grey shading indicates the interannual standard deviation in the catchment-balance approach.

Finally, we assessed interannual trends in Amazon ET over the common time period of 2003–2013, and using all years of available data for each dataset (Table 3). No statistically significant temporal trends were observed for annual, JFM or JAS catchment-balance ET over the respective periods analysed. Removal of the anomalous El Niño year had no impact on the results. Previous studies based on the P-LSH satellite product (Zhang et al., 2015a), and other satellite ET products and machine-learning approaches (Zhang et al., 2016b, Pan et al., 2020) have reported multi-decadal increases in ET, globally and over the Amazon, from the early 1980s to the early 2010s, due to long-term warming driving increased evaporative demand. Meanwhile, climate models predict that Amazon ET will decrease over the next century due to reductions in plant stomatal conductance driven by rising atmospheric $CO_2$ (i.e. the $CO_2$ fertilisation effect), leading to declines in Amazon rainfall (Skinner et al., 2017, Kooperman et al., 2018, Langenbrunner et al., 2019). Swann and Koven (2017) had observed a statistically-significant reduction in monthly catchment-balance Amazon ET from 2002–2016 (–0.12 mm month$^{-1}$ year$^{-1}$), which they hypothesised may have been driven by a reduction in Amazon precipitation, deforestation, or $CO_2$ fertilisation. Our catchment-balance ET data, analysed over a similar period but at the annual timescale, gave a similar value (i.e. –0.09 mm month$^{-1}$ year$^{-1}$, 2003–2013, Table 3), though the result was not statistically significant due to the short length of the time series. Extension of the record to 2019 gave similar a result (Table 3). The absence of a discernible trend in catchment-balance ET in this study suggests that previously reported positive trends in Amazon ET may have levelled off, but that there has not yet been a systematic shift towards long-term reductions in ET driven by precipitation, deforestation or the $CO_2$ fertilisation effect, over the portion of the Amazon drained by Óbidos (Fig. 1), with the caveat that ET changes over the eastern portion of the basin would not be detected in our approach.

Among other datasets, there was little agreement in the direction of ET trends, with both positive and negative trends detected at the annual timescale (P-LSH and MODIS, respectively), and only one product showing a statistically significant upward trend in JAS ET (P-LSH, Table 3). There was more agreement in JFM, with MODIS, GLEAM and ERA5 all showing modest declines in ET (–0.27 to –1.3 mm month$^{-1}$ year$^{-1}$, variable time periods, Table 3). Divergent trends in remote-sensing ET products have been reported previously (Wu et al., 2020). Trends in satellite-derived climate datasets can occur from gradual changes in the satellite orbit over time (drift), which could explain some of the observed trend disparities, although such artefacts should have been corrected for during data processing (Gutman, 1999, Pinzón et al., 2005). Overall, the inconsistencies between satellite, reanalysis and climate model ET records at the interannual timescale, and poor correspondence with catchment-balance ET, highlight that current products are inadequate for evaluating long-term changes in Amazon ET.

## 4 Summary and conclusions

This study aimed to collate estimates of Amazon ET from water-budget analysis, remote sensing, reanalysis, flux-tower measurements and coupled climate models to identify key characteristics of the regional hydrological cycle, compare and evaluate datasets, and identify remaining gaps in our understanding of this important variable. Our quantification of Amazon ET from terms in the water-budget equation revealed a clear spatial gradient in annual ET from east to west/southwest across the Amazon, consistent with measurements from flux towers. We observed a robust seasonal cycle in Amazon-wide ET

peaking in August–October, and no evidence of a long-term trend in annual, January–March or July–September ET from 2001 to 2019. Spatial, seasonal and (to a lesser degree) interannual variation in ET was shown to largely be governed by surface radiation and LAI, highlighting the main factors controlling surface water fluxes in the Amazon region.

The catchment-balance approach, although providing a relatively direct measure of ET, still has a degree of associated

uncertainty (Table S2), and assumes complete closure of the water budget. In particular, subsurface runoff to other catchments and anthropogenic hydrological management could potentially impact the $R$ term in equation 1 (Miralles et al., 2016). Incorporating groundwater measurements, as applied here, should account for sub-surface runoff. However, human encroachment on the Amazon hydrological regime has risen in recent decades with the expansion of hydropower impacting river flow patterns and flood pulse frequency (Fearnside, 2014, Timpe and Kaplan, 2017). ET estimates for the Aripuanã and

whole-Amazon river catchments may have been affected by dam development, though our focus on temporal means made it less likely that our findings were affected by human-induced perturbations to monthly river flows. Furthermore, the generally good agreement between our results and those from previous studies using different data inputs (e.g. Swann and Koven, 2017) provides confidence that our approach was robust.

Performance of satellite, reanalysis and climate model ET was highly variable, though all products overestimated ET at the annual scale (15–37 %), while substantially underestimating temporal variability relative to catchment-balance. In general, satellite ET estimates based on the Penman-Monteith equation (MODIS and P-LSH) showed the best correspondence with catchment-balance ET, mostly capturing spatial and seasonal patterns of variation. The satellite-based GLEAM ET product showed strong positive relationships with rainfall even over very wet parts of the Amazon, suggesting an over-dependence on

soil moisture in the GLEAM land-surface model. ERA5 reanalysis ET performed well at the seasonal scale, and mostly captured the correct relationships with factors controlling ET. However, misrepresentation of other reanalysis variables, including the spatial distribution of precipitation over the Amazon, detrimentally affected ERA5 ET. Our analysis provided a first assessment of Amazon ET representation in the CMIP6 climate models, showing they struggled to capture major features of Amazon ET, including spatial and seasonal variability across the Amazon basin. Furthermore, CMIP6, which represents the

latest generation of coupled climate models, showed little evidence of improvement in the representation of Amazon ET compared to CMIP5, highlighting the need for further process-based model development. It has been suggested that errors in

model rooting (Pan et al., 2020) could play a role in the mischaracterisation of simulated Amazon ET, highlighting a possible area for future research.

Correspondence between ET products at the interannual timescale was particularly poor, suggesting they are currently inadequate for monitoring long-term trends in Amazon ET. Given that changes in ET have implications for regional climate and the sustainability of the Amazon forest biome, there is a clear need for further long-term ground measurements of ET in the region, including direct measures such sap-flow. Although it remains a challenge to scale ground-based ET observations from a few kilometres up to the catchment level of thousands of kilometres, recent advances, such as the installation of the

Amazon Tall Tower Observatory (ATTO), which captures regional processes over a footprint on the order of a thousand kilometres (Andreae et al., 2015), are expected to provide new insights in the field.

The future of Amazon ET is entwined with the fate of the Amazon rainforest, with its rich biodiversity and valuable stores of terrestrial carbon (Malhi et al., 2008, Zhang et al., 2015b). However, uncertainty remains over the direction of future ET trends,

with climate warming and increasing LAI promoting ET increases (Kergoat et al., 2002, Zhang et al., 2015a), and deforestation and $CO_2$-induced reductions in sap flow forcing declines in ET (Zemp et al., 2017a, Skinner et al., 2017, Baker and Spracklen, 2019). Discrepancies in the direction of trends from different ET products in this study make it difficult to assess which of these opposing mechanisms are in operation. Furthermore, deficiencies in the representation of ET in CMIP5 and CMIP6 models highlighted here raise questions over the reliability of Amazon ET projections over the next century, with implications

for other regions. Until models are better able to capture historical patterns of ET and its controlling variables, attempts to understand future changes in the Amazon hydrological cycle will be severely hampered.

## 5 Data and code availability

The observational, reanalysis, model and flux-tower datasets analysed in the study are available in the following repositories:
- CHIRPS precipitation: https://www.chc.ucsb.edu/data/chirps
- Amazon river-gauge station data: http://www2.ana.gov.br/Paginas/EN/default.aspx
- GRACE terrestrial water storage: https://grace.jpl.nasa.gov/data/get-data/jpl_global_mascons/
- MODIS ET: https://lpdaac.usgs.gov/products/mod16a2v006/
- P-LSH ET: https://www.ntsg.umt.edu/project/global-et.php
- GLEAM ET: https://www.gleam.eu
- ERA5 reanalysis: https://climate.copernicus.eu/climate-reanalysis
- CMIP5 and CMIP6 historical simulations: https://esgf-data.dkrz.de/projects/esgf-dkrz/
- CLARA-A1 radiation: https://wui.cmsaf.eu/safira/action/viewDoiDetails?acronym=CLARA_AVHRR_V001
- MODIS leaf area index: https://lpdaac.usgs.gov/products/mod15a2hv006/

- LBA-ECO flux tower data: https://daac.ornl.gov/LBA/guides/CD32_Brazil_Flux_Network.html


We have uploaded a dataset containing Amazon catchment-scale estimates of ET, precipitation, surface radiation and LAI for 2003–2013 from the data sources described in this study to an online repository (https://zenodo.org/record/4271331#.YD-emS2l30o). Catchment-balance error estimates for Amazon ET are also provided (https://zenodo.org/record/4580292#.YECzi2l23c). The scripts used to process the raw data and conduct the catchment-balance analysis are available here:

https://zenodo.org/record/4580447#.YEC_wi2l1hE.

## 6 Author contributions

JB, LGC, MG, JM, WB and DS devised the study, planned the analysis and discussed the results. HR, AN and AA provided ET data, and expertise on Amazon flux-tower measurements. JB performed the analysis and wrote the paper. All authors provided feedback on the manuscript.

**7 Competing interests**

The authors declare that they have no conflict of interest.

## 8 Acknowledgements

The research has been supported by funding from the European Research Council (ERC) under the European Union's Horizon 2020 research and innovation programme (DECAF project, Grant agreement no. 771492), a Natural Environment Research
Council standard grant (NE/K01353X/1), and the Newton Fund, through the Met Office Climate Science for Service Partnership Brazil (CSSP Brazil). The authors also thank contributors to the LBA-ECO Brazil Flux Tower Network for their valuable efforts measuring evapotranspiration in the Amazon and for sharing their data with the scientific community.

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
