# Peer review of "Evapotranspiration in the Amazon: spatial patterns, seasonality and recent trends in observations, reanalysis and CMIP models"

_Hydrology and Earth System Sciences, 2020_

## Referee Comment (RC1) · Anonymous Referee #1 · 25 Dec 2020

This article presents original estimates of evapotranspiration (ET) derived via water balance in the Amazon and several of its subbasins. The authors use these estimates to examine spatial variability in ET within the Amazon, seasonal and interannual variability in ET at the subbasin and Amazon-scale, and the dominant drivers of ET variability in the region. The authors additionally compare their findings to existing ET data from an extensive suite of remotely-sensed ET products, site-scale ET from flux towers, reanalysis models, and CMIP GCMs to evaluate how well they capture the dynamics of Amazonian ET.

I commend the authors for writing a clear and detailed article that will be of interest to

both climate modeling and tropical ecohydrology audiences. Their references to prior literature are thorough and concisely summarized, their findings are presented in well-designed figures, and the details included in the supplementary information will allow readers to build a deep understanding of their analysis. I am also happy to see a clear explanation of error propagation in the paper, which is important for any study using a water balance approach as a benchmark for more complex models.

I do not think any major revisions or additional analyses are needed, but I have some comments and questions listed below that might improve the paper:

L24: The abstract reports a "strong seasonal cycle in basin-mean ET controlled by net incoming radiation...". Here and elsewhere throughout the paper I would consider slightly softening the language, e.g. by saying that basin-wide ET is "primarily controlled by radiation" or "highly correlated with radiation" or something similar. My concern is that while the current statement is consistent with your findings, it is very general and may mislead unfamiliar readers into neglecting other important dynamics behind Amazonian ET. As your results and many other papers have found (Maeda et al. 2017 in ESD comes to mind), vegetation dynamics and water availability (via rainfall as well as terrestrial availability) appear to modulate the ET signal in certain regions during certain times of the year. You have demonstrated quite well that, when aggregated across the entire basin, ET and Rnet have strikingly similar seasonal cycles. But I would be cautious about unintentionally implying radiation is the sole control on ET's seasonality when the full story is more complicated.

L55: The comma after ET seems unnecessary to me.

L199: There appears to be some formatting mistake with the uncertainty term here.

L227: If pasture is the dominant regional land use, why exclude these tower sites? Is pasture just not common enough across your subbasins to be reflected in your tower analysis?

L234: Suggest removing the "to" in "near to Manaus"

L249: It's worth including the version of MOD16A2 that you used, presumably version 6. I would say the same thing for P-LSH and CHIRPS if there are publicized version numbers associated with the data (I think CHIRPS is on version 2 these days).

L279: Why re-grid to 1x1 degree pixels for the visualizations? To reduce noise? I would mention somewhere what the native resolutions of the CMIP models are before resampling.

L295: How many of these gaps were there? If I'm understanding this correctly, each time series is only 11 datapoints, so filling any of those datapoints with climatological means could obscure trends quite a bit. Worth mentioning somewhere in the text or Supplementary Info how much gap-filling was necessary for each basin's ET time series.

L304: On first read I was a bit confused by the term "corresponding sources." In fact this whole paragraph could use some small tweaks for clarity. This reordering of the sentences (and complete removal of the "corresponding sources" sentence) sounds better to me, but you have already demonstrated that you are a capable writer so I will trust you to make whatever changes you see fit:

"...Bins with fewer than five data points were excluded from the analysis. Satellite-based ET estimates were binned according to precipitation from CHIRPS, radiation from CLARA-A1 (Karlsson et al., 2013) and LAI from the MODIS MOD15A2H product (Myneni et al., 2015), each re-gridded to $0.25°$ x $0.25°$, while reanalysis and model ET were compared with reanalysis and model variables, respectively. For ERA5, we used the 'high vegetation' LAI field since the Amazon is predominantly covered with tropical forest, though repeating the analysis with 'low vegetation' LAI made little difference to the results. Note that the satellite-based MODIS ET..."

L306: I would appreciate some very brief discussion of how trustworthy CLARA and

MOD15A2 can be considered in this region, if anything is known. For instance, does MOD15A2 suffer from the sun-sensor geometry issues known to affect MODIS in the Amazon? I don't know much about the CLARA-A1 dataset but perhaps there is some validation study available, or at least you could explain why you chose it over other radiation datasets.

L310: Was the Amazon's hydroclimate during 1994-2004 broadly similar to 2003-2013? I'm not immediately familiar with the Amazon's recent climate trends, but if one period had worse droughts or wetter wet years than the other, mean ET may show a response to that. Somewhere in the paper, one or two lines covering this question might be a good addition.

Figure 3: Is there some kind of interpolation or smoothing being used here? My understanding was that the CMIP models were resampled to 1-degree grid cells for visualization, which seems much coarser than the data in panels g and h. I don't necessarily take issue with interpolating for the presentation of this figure, but it may provide a false sense of spatial detail and should be explicitly stated somewhere.

L354: I don't know if I'd say the tower gradient is "similar" given so few datapoints, and the two northeastern towers featuring such different values. Perhaps just say they appear to display an east-west gradient. Some explanation of why the two very close towers feature quite different mean ETs would be welcome as well—is it a land cover difference?

L377: As I wrote regarding the abstract, I would be careful with statements like "water availability is not a limiting factor controlling the spatial distribution of ET over the Amazon." It seems to me that P may well limit ET in some catchments (e.g. Madeira, which has relatively low ET and P but relatively high RDN in your plots. Maeda et al. (2017) characterized Madeira and other basins as water-limited for at least part of the year). I think you could justifiably conclude "water availability does not consistently limit ET in all regions of the Amazon" from Figure 4, or that "water availability does not limit ET as

consistently across Amazonian subbasins as radiation." But as it is now, I think the line is potentially misleading.

Figure 4 (and others): The symbols for Amazon and Tapajos are difficult to distinguish without zooming in. Fortunately you have prepared your figures well enough that zooming in is possible, but I worry that people reading printed versions of the article would still have trouble. Maybe Tapajos could be replaced with an unfilled circle?

Figure 7: I am curious why you decided to do this analysis on monthly climatological means rather than just plotting LAI and ET in every individual month you had available. Also, it looks like you are calling P-LSH "LSE-Zhang" in the plot titles; I would change this to be consistent throughout.

Supplemental Information:

Table S2: Please clarify what area these values pertain to.

Table S6: Change Zhang to P-LSH?

Figure S4: In last sentence of the caption, change "was" to "were."

Figure S7: I would love to see this figure replicated for RDN and LAI, since so much of your analysis of various ET data sources hinges on the comparisons to their respective RDN and LAI datasets. After reading your article I was left wondering how spatially variable these drivers are in the reanalysis and GCMs.

Figure S9: If it's not too much trouble, a similar plot showing LAI's seasonal cycle across the basin would be interesting since LAI appears to relate to ET when viewed across subbasins and months. Adding to this figure may make it too crowded, so perhaps just in a new supplemental figure.

Figure S10: I don't understand what the colors in the caption are referring to. Where is dark blue? What about red and magenta?
[Figure]

---

## Referee Comment (RC2) · Anonymous Referee #2 · 25 Jan 2021

This work is generally well-written and addresses an important topic: hydrometeorology/hydroclimatology in the Amazon. The methodology is sound and very well explained. I particularly liked the discussion about errors estimated for the catchment-based ET estimates. The idea is to compare those estimates against a number of other sources including satellite-based products, reanalysis, and CMIP5/CMIP6 model outputs for the region. I believe this paper will be a good addition to HESS and I only have some minor comments to the authors (in no specific order of importance):

1. The abstract ends with a recommendation for the need for more ground based ET observations. If that is the case, I suggest the authors to expand more on that in the
discussion including challenges, especially those associated with spatial scaling of ET flux tower estimates to catchment-/basin-wide estimates.

2. Note Barlow et al. 2020 reference is not provided

3. The addition of GRACE as the ds/dt term and propagation of error was very nicely included. Just a comment

4. Figure 2 (data analysis in general): Have the authors considered comparing the PDFs of those? Perhaps apply Kolmogorov-Smirnov test to check whether these series come (or not) from the same distribution? Assuming this can be done at highest common temporal resolution possible among different data sources (monthly???).

5. Figures 2 and 3 (and in general): Have the authors masked out the regions from the satellite and model products where P-R and catchment were not computed, to ensure direct comparison?

6. Figure 4 and 7: How much confidence on those statistics and ultimately interpretation of results with too fewer points? Can the authors expand this discussion and implications?

---

## Author Comment (AC1) · 31 Jan 2021

This article presents original estimates of evapotranspiration (ET) derived via water balance in the Amazon and several of its subbasins. The authors use these estimates to examine spatial variability in ET within the Amazon, seasonal and interannual variability in ET at the subbasin and Amazon-scale, and the dominant drivers of ET variability in the region. The authors additionally compare their findings to existing ET data from an extensive suite of remotely-sensed ET products, site-scale ET from flux towers, reanalysis models, and CMIP GCMs to evaluate how well they capture the dynamics of Amazonian ET.

I commend the authors for writing a clear and detailed article that will be of interest to both climate modeling and tropical ecohydrology audiences. Their references to prior literature are thorough and concisely summarized, their findings are presented in well-designed figures, and the details included in the supplementary information will allow readers to build a deep understanding of their analysis. I am also happy to see a clear explanation of error propagation in the paper, which is important for any study using a water balance approach as a benchmark for more complex models.

Answer: We would like to sincerely thank you for taking the time to carefully read and review our work. We have revised the manuscript according to your suggestions, and feel that is now a stronger paper as a result. We respond to each of your comments in the text below.

I do not think any major revisions or additional analyses are needed, but I have some comments and questions listed below that might improve the paper:

L24: The abstract reports a "strong seasonal cycle in basin-mean ET controlled by net incoming radiation...". Here and elsewhere throughout the paper I would consider slightly softening the language, e.g. by saying that basin-wide ET is "primarily controlled by radiation" or "highly correlated with radiation" or something similar. My concern is that while the current statement is consistent with your findings, it is very general and may mislead unfamiliar readers into neglecting other important dynamics behind Amazonian ET. As your results and many other papers have found (Maeda et al. 2017 in ESD comes to mind), vegetation dynamics and water availability (via rainfall as well as terrestrial availability) appear to modulate the ET signal in certain regions during certain times of the year. You have demonstrated quite well that, when aggregated across the entire basin, ET and Rnet have strikingly similar seasonal cycles. But I would be cautious about unintentionally implying radiation is the sole control on ET's seasonality when the full story is more complicated.

Answer: This is a good point, and we have now amended the text to be more careful

in our language when describing the controls on ET.

L55: The comma after ET seems unnecessary to me.

Answer: We have removed the comma.

L199: There appears to be some formatting mistake with the uncertainty term here.

Answer: We have corrected the typo.

L227: If pasture is the dominant regional land use, why exclude these tower sites? Is pasture just not common enough across your subbasins to be reflected in your tower analysis?

Answer: We apologise for miswording this sentence. The pasture towers we excluded were in areas where the dominant land cover was forest, rather than pasture, and thus the towers were not representative of the surrounding land cover. We have corrected the text.

L234: Suggest removing the "to" in "near to Manaus"

Answer: Corrected.

L249: It's worth including the version of MOD16A2 that you used, presumably version 6. I would say the same thing for P-LSH and CHIRPS if there are publicized version numbers associated with the data (I think CHIRPS is on version 2 these days).

Answer: Thanks for this suggestion. We have added the version numbers for MOD16A2 and CHIRPS to the text. To our understanding, there is no published version number associated with the P-LSH dataset.

L279: Why re-grid to 1x1 degree pixels for the visualizations? To reduce noise? I would mention somewhere what the native resolutions of the CMIP models are before resampling.

Answer: The CMIP models vary from model to model in their resolution, and in order to

calculate a multi-model mean it was necessary to harmonise the models to a consistent resolution. We selected 1° x 1°, which is towards the finer end of the model resolutions, as this enabled us to extract data from each Amazon sub-basin with more accuracy than with a coarser grid. We have added columns to Tables S3 & S4 giving the native resolution of each model and added a line to the text emphasising that though not all models simulate the level of detail provided by 1° x 1°, choosing this resolution enabled us to extract data from each Amazon sub-basin with more accuracy than using a coarser grid.

L295: How many of these gaps were there? If I'm understanding this correctly, each time series is only 11 datapoints, so filling any of those datapoints with climatological means could obscure trends quite a bit. Worth mentioning somewhere in the text or Supplementary Info how much gap-filling was necessary for each basin's ET time series.

Answer: Sorry this sentence was included in error. We calculated interannual trends for the Amazon basin only and had originally used a gap-filling approach for the catchment-balance series, but on reflection decided to exclude years with fewer than 10 months of data, or any years with a missing month for the JFM and JAS time series. Therefore, years 2017 and 2018 were removed from the annual time series (Fig. 8a), and the year 2017 from the JAS time series (Fig. 8c). The trends calculated for 2003–2013 were unaffected by data missing from these years. We have amended the text to clarify and correct the description of the methods.

L304: On first read I was a bit confused by the term "corresponding sources." In fact this whole paragraph could use some small tweaks for clarity. This reordering of the sentences (and complete removal of the "corresponding sources" sentence) sounds better to me, but you have already demonstrated that you are a capable writer so I will trust you to make whatever changes you see fit: ". . .Bins with fewer than five data points were excluded from the analysis. Satellite- based ET estimates were binned according to precipitation from CHIRPS, radiation from CLARA-A1 (Karlsson et al., 2013)

[Figure]

and LAI from the MODIS MOD15A2H product (Myneni et al., 2015), each re-gridded to 0.25ậŮę x 0.25ậŮę, while reanalysis and model ET were compared with reanalysis and model variables, respectively. For ERA5, we used the 'high vegetation' LAI field since the Amazon is predominantly covered with tropical forest, though repeating the analysis with 'low vegetation' LAI made little difference to the results. Note that the satellite-based MODIS ET. . ."

Answer: Many thanks for this helpful suggestion to improve the clarity of the text. We have amended the paragraph in line with your suggestions and agree that it now reads more clearly.

L306: I would appreciate some very brief discussion of how trustworthy CLARA and MOD15A2 can be considered in this region, if anything is known. For instance, does MOD15A2 suffer from the sun-sensor geometry issues known to affect MODIS in the Amazon? I don't know much about the CLARA-A1 dataset but perhaps there is some validation study available, or at least you could explain why you chose it over other radiation datasets.

Answer: This is an important point, and we have added some discussion on the validity of these products over the Amazon to section 2.6:

"MODIS LAI has been shown to perform relatively well against ground-based LAI measurements (R2=0.7–0.77), though uncertainty over the validity of high LAI values (>4 m2 m-2), such as occur over the Amazon, is larger due there being few ground measurements and the satellite reflectance signal reaching saturation over dense canopies (Yan et al., 2016a). Furthermore, the satellite-based MODIS ET product incorporates MODIS LAI (Table 1), and thus these datasets are not fully independent from one another. The CLARA-A1 radiation is independent from the ET datasets evaluated in this study and estimated to have an accuracy of ≤10 W m-2, though few surface measurements were available over South America, and none in the Amazon region (Karlsson et al., 2013). Thus, there is some uncertainty in the accuracy of these satellite products

over the Amazon that must be considered when interpreting the results."

L310: Was the Amazon's hydroclimate during 1994-2004 broadly similar to 2003-2013? I'm not immediately familiar with the Amazon's recent climate trends, but if one period had worse droughts or wetter wet years than the other, mean ET may show a response to that. Somewhere in the paper, one or two lines covering this question might be a good addition.

Answer: You are right that there has been a change in the Amazon hydroclimate between these two periods, with the Amazon hydrological cycle becoming more intense and basin-mean P increasing (Gloor et al., 2013). Therefore, we might expect to see some differences between CMIP5 results and results from other data sources. However, we found CMIP5 ET to be largely consistent with ET from CMIP6, despite the different time periods analysed, suggesting differences due to time period were smaller than differences between the models and other types of ET data. We have added a comment on this to section 2.6.

Figure 3: Is there some kind of interpolation or smoothing being used here? My understanding was that the CMIP models were resampled to 1-degree grid cells for visualization, which seems much coarser than the data in panels g and h. I don't necessarily take issue with interpolating for the presentation of this figure, but it may provide a false sense of spatial detail and should be explicitly stated somewhere.

Answer: You are right, we used the filled-contour plot function to map the data, with contours at 25-mm intervals between 1000 and 1500 mm year-1. We have now added this information to the figure caption for transparency.

L354: I don't know if I'd say the tower gradient is "similar" given so few datapoints, and the two northeastern towers featuring such different values. Perhaps just say they appear to display an east-west gradient. Some explanation of why the two very close towers feature quite different mean ETs would be welcome as well – is it a land cover difference?

Answer: We have amended the text to make a more careful comment on the possible gradient suggested by the flux tower data. We have also added a statement stating that the observable difference in mean annual ET between the two nearby towers in the northeast amazon is likely due to these being on different land-cover types (primary forest and selectively logged forest).

L377: As I wrote regarding the abstract, I would be careful with statements like "water availability is not a limiting factor controlling the spatial distribution of ET over the Amazon." It seems to me that P may well limit ET in some catchments (e.g. Madeira, which has relatively low ET and P but relatively high RDN in your plots. Maeda et al. (2017) characterized Madeira and other basins as water-limited for at least part of the year). I think you could justifiably conclude "water availability does not consistently limit ET in all regions of the Amazon" from Figure 4, or that "water availability does not limit ET as consistently across Amazonian subbasins as radiation." But as it is now, I think the line is potentially misleading.

Answer: Thanks for drawing our attention to this ambiguously-worded sentence. We did not mean to give the impression that water-limitation never occurs over the Amazon, rather that spatial variation in ET across the Amazon appeared to be associated with spatial variation in radiation, and not water availability. We have rewritten the sentence in question and made sure to be more careful with our language: "This result tentatively suggests that spatial variation in radiation explains more of the spatial variability in ET across Amazon sub-catchments than other variables." The text of the whole paragraph is copied at the bottom of this document.

Figure 4 (and others): The symbols for Amazon and Tapajos are difficult to distinguish without zooming in. Fortunately you have prepared your figures well enough that zooming in is possible, but I worry that people reading printed versions of the article would still have trouble. Maybe Tapajos could be replaced with an unfilled circle?

Answer: We have amended the symbol for Tapajos in Figures 4, 5, S1 and S7 to a star

[Figure]

which is more distinguishable from the circle than the hexagon, and increased the size of the markers.

Figure 7: I am curious why you decided to do this analysis on monthly climatological means rather than just plotting LAI and ET in every individual month you had available. Also, it looks like you are calling P-LSH "LSE-Zhang" in the plot titles; I would change this to be consistent throughout.

Answer: We wanted to separate the controls on seasonal variation in ET (Fig. 7) from the controls on interannual variation in ET (Fig. S10), which is why we only plotted the climatological monthly means in this plot. We have amended the Method to clarify our reasoning. We have corrected the title for the P-LSH panels – thanks for drawing our attention to this error.

Supplemental Information:

Table S2: Please clarify what area these values pertain to.

Answer: We have added this information.

Table S6: Change Zhang to P-LSH?

Answer: Correction made.

Figure S4: In last sentence of the caption, change "was" to "were."

Answer: Correction made.

Figure S7: I would love to see this figure replicated for RDN and LAI, since so much of your analysis of various ET data sources hinges on the comparisons to their respective RDN and LAI datasets. After reading your article I was left wondering how spatially variable these drivers are in the reanalysis and GCMs.

Answer: This is a very nice idea – thank you for the suggestion. We have expanded the figure to include maps showing climatological annual mean radiation and leaf area

index, in addition to precipitation. These maps (attached Figure 1) clearly show that radiation is higher in the models than in the satellite or reanalysis products, particularly in the eastern Amazon. Satellite LAI also shows lower variability than other products, likely due to signal saturation. We have added this to the discussion.

Figure S9: If it's not too much trouble, a similar plot showing LAI's seasonal cycle across the basin would be interesting since LAI appears to relate to ET when viewed across subbasins and months. Adding to this figure may make it too crowded, so perhaps just in a new supplemental figure.

Answer: Thank you for this useful suggestion. We have added a figure showing the satellite LAI and catchment-balance ET seasonal cycles side by side (see attached Figure 2) and refer to it in the text.

Figure S10: I don't understand what the colors in the caption are referring to. Where is dark blue? What about red and magenta?

Answer: We have corrected this figure caption (now Fig. S12) and removed the references to colours. Apologies for this error which related to an earlier version of the figure.

Other changes

We noticed that our Amazon LAI values were implausibly low (Amazon mean LAI value of 3.6 m2/m2), likely due to inadequate quality control during data processing. We have changed to use a quality-controlled MODIS MOD15A2H Collection 6 LAI dataset provided by Boston University (Amazon mean LAI value of 4.4 m2/m2). The main difference to the results arising from this change is that catchment-balance ET is no longer well related to spatial variation in LAI (see attached Figure 3). The new figure and paragraph describing these results are copied below. There were no meaningful changes to any of the rest of the results.

"To understand the drivers of spatial variation in Amazon ET, we compared catchmentscale estimates against catchment-means of precipitation, surface radiation and LAI (Fig. 4). Since there were only eleven data points in the analysis (representing the Amazon and ten sub-catchments), statistical power was relatively low. However, we found spatial variation in catchment-balance ET showed some indication of an influence from radiation (r=0.38, p=0.25, Fig. 4h), but not precipitation (r=0.14, p=0.68, Fig. 4a) or LAI (r=0.06, p=0.87, Fig. 4o). This result tentatively suggests that spatial variation in radiation explains more of the spatial variability in ET across Amazon sub-catchments than other variables. None of the ET products and models analysed captured positive relationships between catchment-mean ET and radiation. ET from ERA5 and the CMIP ensembles instead showed negative associations with radiation (Fig. 4l–n), and, along with GLEAM ET, positive relationships with precipitation (Fig. 4d–g), indicative of water availability influencing spatial variation in ET (Fig. 4d–g). These results confirm that the reanalysis and climate models analysed here struggled to capture spatial patterns in Amazon ET due to misrepresentation of the controlling drivers, specifically the relative importance of precipitation and net radiation. ET from ERA5 and the models also showed positive correlations between LAI and ET (Fig. 4s–u), not seen in the satellite observations. However, it should be noted that satellite LAI was generally lower and showed less spatial variability than other LAI datasets over the Amazon (Fig. S8i–l), likely due to the satellite sensor being insensitive to variation in LAI over areas of dense tropical forest (Myneni et al., 2002, Yan et al., 2016a). This could hamper our ability to accurately assess the extent to which LAI influences spatial variation in ET."

References

Gloor, M., Brienen, R. J. W., Galbraith, D., Feldpausch, T. R., Schöngart, J., Guyot, J. L., Espinoza, J. C., Lloyd, J. & Phillips, O. L. 2013. Intensification of the Amazon hydrological cycle over the last two decades. Geophysical Research Letters, 40, 1729-1733.

Please also note the supplement to this comment:
https://hess.copernicus.org/preprints/hess-2020-523/hess-2020-523-AC1-supplement.pdf

———————————————————

[Figure]

**Fig. 1.** Climatological annual precipitation, radiation and leaf area index. Mean annual precipitation (P, a–d), radiation (RDN, e–h) and leaf area index (LAI, i–l) from satellites (column 1), reanalysis (colu

[Figure]

**Fig. 2.** Seasonal variation in leaf area index over the Amazon. Climatological seasonal cycles in catchment-balance ET (blue) and MODIS MOD15A2H Collection 6 LAI (green) averaged over the Amazon region shown i

[Figure]

[Figure]

**Fig. 3.** Controls on spatial variation in Amazon evapotranspiration. Annual mean ET (in mm month-1) for the Amazon and ten sub-catchments (Fig. 1) from catchment-balance, satellites (MODIS, P-LSH, G

---

## Author Comment (AC2) · 31 Jan 2021

This work is generally well-written and addresses an important topic: hydrometeorology/hydroclimatology in the Amazon. The methodology is sound and very well explained. I particularly liked the discussion about errors estimated for the catchment-based ET estimates. The idea is to compare those estimates against a number of other sources including satellite-based products, reanalysis, and CMIP5/CMIP6 model outputs for the region.

We would like to sincerely thank you for taking the time to read our paper and make suggestions for its improvement. We have revised the manuscript according to your

suggestions, and feel that is now a stronger paper as a result. We respond to each of your comments in the text below.

I believe this paper will be a good addition to HESS and I only have some minor comments to the authors (in no specific order of importance):

1. The abstract ends with a recommendation for the need for more ground based ET observations. If that is the case, I suggest the authors to expand more on that in the discussion including challenges, especially those associated with spatial scaling of ET flux tower estimates to catchment-/basin-wide estimates.

Answer: This is a good point, because although we highlighted the need for more ground-based observations in the abstract, we had not fully discussed the implications of this, or mentioned the challenges of comparing point-based ET measurements with grid-cell level estimates. We have expanded the discussion describing these issues, and possible solutions to overcome them. For example, the establishment of the Amazon Tall Tower Observatory in 2015, which has a footprint on the order of a thousand kilometres (compared to just a few kilometres for conventional towers), may offer a way to monitor ET over scales that are more directly comparable with coarse-scale models and satellite products.

2. Note Barlow et al. 2020 reference is not provided

Answer: Thanks for drawing this to our attention – we have added in the reference.

3. The addition of GRACE as the ds/dt term and propagation of error was very nicely included. Just a comment

Answer: We thank the reviewer for this comment on our work.

4. Figure 2 (data analysis in general): Have the authors considered comparing the PDFs of those? Perhaps apply Kolmogorov-Smirnov test to check whether these series come (or not) from the same distribution? Assuming this can be done at highest common temporal resolution possible among different data sources (monthly???).

Answer: Thanks for this interesting and useful suggestion. We calculated the two-sample Kolmogorov-Smirnov statistic to identify whether monthly Amazon ET values from 2003 to 2013 from satellite, reanalysis and climate models were drawn from the same distribution as the catchment-balance ET values. All of the ET datasets that we analysed were shown to be from statistically different distributions to the catchment-balance data. We plotted the cumulative probability and probability density functions, as shown below. ET products and models show much narrower distributions of ET, and miss the low values present in the catchment-balance data. This additional analysis helps to further explain why climatological Amazon annual mean ET is overestimated (see attached Figure 1). We have added this figure to the Supplementary Material (Fig. S5).

Caption Figure 1– Kolmogorov-Smirnov analysis. The cumulative probability (a) and probability density (b) functions for monthly Amazon ET from catchment-balance, satellites (MODIS, P-LSH, GLEAM), ERA5 reanalysis, and climate models (CMIP5 and CMIP6) from 2003 to 2013. Dashed lines indicate the data come from a statistically different distribution from the catchment-balance data (determined using a two-sample Kolmogorov-Smirnov test).

5. Figures 2 and 3 (and in general): Have the authors masked out the regions from the satellite and model products where P-R and catchment were not computed, to ensure direct comparison?

Answer: Yes, when comparing catchment-balance ET and data extracted from gridded products we only analysed data over the region drained by Obidos (indicated by blue hatching in Fig. 1 and on the inset map in Fig. 2). We have added clarification to the figure caption.

6. Figure 4 and 7: How much confidence on those statistics and ultimately interpretation of results with too fewer points? Can the authors expand this discussion and implications?

Answer: The small sample sizes were partly due to our decision to separate out the seasonal signal from the interannual signal in ET, as we felt that this could potentially provide more useful information about controls on ET over the Amazon. But admittedly a drawback of this approach was that statistical power was correspondingly low. This means when we did not detect a statistically significant signal then it could either be because there was no signal to detect, or because the signal-to-noise ratio was too low. For the seasonal analysis, we did find statistically significant relationships between catchment-balance ET and radiation (r=0.93, p<0.001) and between catchment-balance ET and LAI (r=0.63, p<0.05). However, spatial and interannual correlations were generally weaker and found not to be statistically significant. We have revised the Methods and Discussion sections to specifically acknowledge that by focussing our analysis on the satellite era, and the period of overlap with the CMIP models, the time period of evaluation was relatively limited, and that our decision to distinguish between seasonal and interannual controls on ET meant analysing relationships between short time series, which should be taken into consideration when interpreting the results.

Other changes

We noticed that our Amazon LAI values were implausibly low (Amazon mean LAI value of 3.6 m2/m2), likely due to inadequate quality control during data processing. We have changed to use a quality-controlled MODIS MOD15A2H Collection 6 LAI dataset provided by Boston University (Amazon mean LAI value of 4.4 m2/m2). The main difference to the results arising from this change is that catchment-balance ET is no longer well related to spatial variation in LAI. The new figure and paragraph describing these results are copied below (see attached Figure 2). There were no meaningful changes to any of the rest of the results.

"To understand the drivers of spatial variation in Amazon ET, we compared catchment-scale estimates against catchment-means of precipitation, surface radiation and LAI (Fig. 4). Since there were only eleven data points in the analysis (representing the Amazon and ten sub-catchments), statistical power was relatively low. However, we

found spatial variation in catchment-balance ET showed some indication of an influ-ence from radiation (r=0.38, p=0.25, Fig. 4h), but not precipitation (r=0.14, p=0.68, Fig. 4a) or LAI (r=0.06, p=0.87, Fig. 4o). This result tentatively suggests that spa-tial variation in radiation explains more of the spatial variability in ET across Amazon sub-catchments than other variables. None of the ET products and models analysed captured positive relationships between catchment-mean ET and radiation. ET from ERA5 and the CMIP ensembles instead showed negative associations with radiation (Fig. 4l–n), and, along with GLEAM ET, positive relationships with precipitation (Fig. 4d–g), indicative of water availability influencing spatial variation in ET (Fig. 4d–g). These results confirm that the reanalysis and climate models analysed here struggled to capture spatial patterns in Amazon ET due to misrepresentation of the controlling drivers, specifically the relative importance of precipitation and net radiation. ET from ERA5 and the models also showed positive correlations between LAI and ET (Fig. 4s–u), not seen in the satellite observations. However, it should be noted that satellite LAI was generally lower and showed less spatial variability than other LAI datasets over the Amazon (Fig. S8i–l), likely due to the satellite sensor being insensitive to variation in LAI over areas of dense tropical forest (Myneni et al., 2002, Yan et al., 2016a). This could hamper our ability to accurately assess the extent to which LAI influences spatial variation in ET."

Caption Figure 2 – Controls on spatial variation in Amazon evapotranspiration. Annual mean ET (in mm month-1) for the Amazon and ten sub-catchments (Fig. 1) from catchment-balance, satellites (MODIS, P-LSH, GLEAM), ERA5 reanalysis, and climate models (CMIP5 and CMIP6), plotted against (a–g) precipitation (P, mm month-1); (h–n) surface shortwave radiation (RDN, W m-2); and (o–u) leaf area index (LAI, m2 m-2). Satellite ET data are plotted against P from CHIRPS, RDN from CLARA-A1 and LAI from MODIS; ERA5 and climate model ET are plotted against ERA5 and model P, RDN and LAI, respectively. Data are from 2003 to 2013, with the exception of CMIP5, where the data are from 1994–2004. Note that the axes do not start at zero.

Please also note the supplement to this comment:
https://hess.copernicus.org/preprints/hess-2020-523/hess-2020-523-AC2-supplement.pdf

―――――――――――――――――――――

[Figure]

[Figure]

**Fig. 1.** Kolmogorov-Smirnov analysis

[Figure]

**Fig. 2.** Controls on spatial variation in Amazon evapotranspiration

---

## Referee Comment (RC3) · Anonymous Referee #3 · 1 Feb 2021

General Comments: The authors compare the spatial patterns, seasonality, interannual variations, and trends of evapotranspiration (ET) estimates from several satellite products, reanalysis, and climate models with catchment scale mass balance ET and ET estimates from flux towers. ET is the largest flux from the land to the atmosphere. Despite its importance, estimates of ET are often highly uncertain. This uncertainty in ET estimates poses a real challenge in hydrological and land atmospheric studies at several scales. In this regard the authors are trying to address an important open question by analyzing the consistency and reliability of a few remotely sensed products and modeled ET fluxes. To do so, they compared these ET estimates with long-term catchment mass balance ET.

Major comments My major concern is that the uncertainty in catchment mass balance ET that the authors correctly report (Fig 6b and 6c) encapsulates almost all model estimates of ET and remotely sensed ET products. Despite being well written and well structured, I doubt that the basic method of using catchment mass balance ET with such large uncertainty is suitable to evaluate the performance of ET products. Presenting correlations of modeled/estimated ET with catchment mass balance values as performance measures is questionable given the large uncertainties involved in catchment mass balance estimates. Therefore, it is not clear to me in what ways the current paper adds up to the currently reported literature on uncertainties in ET products (e.g. Mueller et al 2014) or in other words what is new here that we didn't know before? For example in Figure 6 b and c, almost all model estimates of ET falling within the uncertainty band of catchment-balance ET. Given such a large uncertainty, one cannot judge the suitability of any ET product.

My second major concern is that given the very coarse resolution of GRACE data (300 km, smoothed to 200 or 100 km), and the dependence of neighboring grid cells, how much GRACE signal adds value to the analysis? How large is the changes in storage as compared to the uncertainties of the other terms in the mass balance eq.?

Specific comments: Line 161-165: How large is the total bias at the spatial scales that is relevant to GRACE observations? Line 185-189: GRACE data contains three observations per month and reported as monthly data, same as runoff and precipitation in the current study. Not sure the need for interpolation here? Line 220, not sure if this arbitrary exclusion of observations is justifiable (specially for the year 2018). Please report (explanation/graph) in what ways including/excluding these points affect your results. Line 226-228: Please explain in what ways inclusion of these sites (pasture) would affect your analysis. Figure 4 and 7: Are these data points extracted for the grid cells where flux towers are located (given the symbols on the plot). In general in most figures it is hard to understand over which spatial scales the ET estimates are aggregated. Figure 8: For all panels please provide the uncertainty band (at least for

catchment mass balance ET). Line 555-560: This is all known and well reported in the literature. What is new? A general comment: Why analysis of long-term monthly values and not presenting monthly data for all products?

---

## Author Comment (AC3) · 9 Feb 2021

General Comments: The authors compare the spatial patterns, seasonality, interannual variations, and trends of evapotranspiration (ET) estimates from several satellite products, reanalysis, and climate models with catchment scale mass balance ET and ET estimates from flux towers. ET is the largest flux from the land to the atmosphere. Despite its importance, estimates of ET are often highly uncertain. This uncertainty in ET estimates poses a real challenge in hydrological and land atmospheric studies at several scales. In this regard the authors are trying to address an important open question by analyzing the consistency and reliability of a few remotely sensed products

and modeled ET fluxes. To do so, they compared these ET estimates with long-term catchment mass balance ET.

Answer: We thank the referee for taking the time to read and critique our paper. We have responded to each of your comments in the text below.

Major comments My major concern is that the uncertainty in catchment mass balance ET that the authors correctly report (Fig 6b and 6c) encapsulates almost all model estimates of ET and remotely sensed ET products. Despite being well written and well structured, I doubt that the basic method of using catchment mass balance ET with such large uncertainty is suitable to evaluate the performance of ET products. Presenting correlations of modeled/estimated ET with catchment mass balance values as performance measures is questionable given the large uncertainties involved in catchment mass balance estimates. Therefore, it is not clear to me in what ways the current paper adds up to the currently reported literature on uncertainties in ET products (e.g. Mueller et al 2014) or in other words what is new here that we didn't know before? For example in Figure 6 b and c, almost all model estimates of ET falling within the uncertainty band of catchment-balance ET. Given such a large uncertainty, one cannot judge the suitability of any ET product.

Answer: We thank the referee for raising uncertainty in the catchment balance approach and the need to consider this uncertainty whilst assessing different ET products. Our main focus in Figure 6 is at the Amazon basin scale (Fig. 6a), where it is clear that the ET products and models fall outside the uncertainty of the catchment balance approach. The robust seasonal cycle in catchment-balance ET is consistent with studies reporting increasing photosynthetic capacity or 'green-up' in the Amazon dry season, when light availability increases. We agree that more care has to be taken when estimating catchment-balance ET at smaller spatial scales, and as such we have decided to move panels 6b and 6c to the Supplementary Material. We have expanded the discussion to acknowledge uncertainties in the catchment-balance approach, particularly when applied over smaller areas, including adding the following statement:

"We note that relative uncertainties in ET estimated using the catchment balance approach increase at smaller spatial scales, precluding a more in-depth assessment of the different ET datasets at smaller scales."

My second major concern is that given the very coarse resolution of GRACE data (300 km, smoothed to 200 or 100 km), and the dependence of neighboring grid cells, how much GRACE signal adds value to the analysis? How large is the changes in storage as compared to the uncertainties of the other terms in the mass balance eq.?

Answer: We thank the reviewer for questions on our inclusion of GRACE data. Data from GRACE is crucial as it allows us to estimate monthly ET from the catchment-balance approach. This is because groundwater storage in the Amazon has a strong seasonal cycle, as shown in Figures S3 & S4. Without the GRACE data, ET estimated as P–R is restricted to multi-annual means. In Figure 2, we compare ET estimated as P–R and ET estimated from catchment balance (including GRACE data) and demonstrate that including GRACE data does not bias the results. We already report uncertainties in ET estimated using catchment balance approach (see Table S2), allowing uncertainties in GRACE to be compared with other components of the water balance. The referee is correct that uncertainty in GRACE data is substantial. However, it is not much larger than uncertainty in the precipitation data (e.g. mean absolute uncertainty is 8.72 mm in GRACE vs. 6.76 mm for CHIRPS precipitation). Furthermore, at larger spatial scales, such as over the whole Amazon, our results were shown to be robust. Overall, we feel that inclusion of GRACE data, alongside a careful consideration of the uncertainties, is important and makes a useful contribution to our analysis.

Specific comments: Line 161-165: How large is the total bias at the spatial scales that is relevant to GRACE observations?

Answer: Error and bias in CHIRPS data is assessed and reported through comparison with in-situ rain gauges [we refer to Paredes-Trejo et al. (2017)]. It is challenging to assess error at larger spatial scales due to the lack of appropriate data at these scales.

[Figure]

For this reason, we report error and bias in CHIRPS as reported previously in the literature.

Line 185-189: GRACE data contains three observations per month and reported as monthly data, same as runoff and precipitation in the current study. Not sure the need for interpolation here?

Answer: The GRACE data that we obtained contained data at irregular intervals (ten or eleven timesteps year) and thus interpolation was required to get monthly-resolution data. The reason for this is related to management of batteries on the GRACE satellites, causing gaps in the GRACE data every 5–6 months for a period of 4–5 weeks. We have added this reasoning to the text.

Line 220, not sure if this arbitrary exclusion of observations is justifiable (specially for the year 2018). Please report (explanation/graph) in what ways including/excluding these points affect your results.

Answer: Including or excluding these data had no effect on the main results reported in the paper, which focussed on the period 2003–2013 (common period across all datasets), but may have influenced trends in ET calculated over the full period. We recalculated the annual, JAS and JFM trends with and without these data points and found no statistically significant trend whether these years were included in the analysis or not. We added the following comment to the text:

"We tested the sensitivity of our interannual trend analysis to the removal of these data points and found it had no statistically significant impact on the reported results."

Line 226-228: Please explain in what ways inclusion of these sites (pasture) would affect your analysis.

Answer: We apologise for miswording this sentence. The pasture towers we excluded were in areas where the dominant land cover was forest, rather than pasture, and thus the towers were not representative of the surrounding land cover. We have corrected

the text.

Figure 4 and 7: Are these data points extracted for the grid cells where flux towers are located (given the symbols on the plot). In general in most figures it is hard to understand over which spatial scales the ET estimates are aggregated.

Answer: Apologies for any confusion. Figure 4 shows the catchment-mean ET estimates for the Amazon and its ten sub-catchments that are shown in Figure 1, plotted against possible controlling variables. In Figure 7, ET is averaged over the area drained by Óbidos, which is also indicated in Figure 1. We indicate this information in the caption of each figure for clarity.

Figure 8: For all panels please provide the uncertainty band (at least for catchment mass balance ET).

Answer: We have now added shading to indicate the uncertainty in catchment-balance ET at the interannual timescale to all panels (see attached Fig. 1).

Line 555-560: This is all known and well reported in the literature. What is new? A general comment: Why analysis of long-term monthly values and not presenting monthly data for all products?

Answer: We thank the referee for pointing out that we had not fully highlighted the novel aspects of our analysis. Importantly, our paper is the first to evaluate ET from CMIP6 models across the Amazon, comparing model output with satellite observations, reanalysis and catchment-balance estimates of ET. We show that the robust seasonal cycle in ET at the Amazon-basin scale is poorly represented by climate models, raising doubts over their ability to simulate future changes in Amazon hydrology. We add the following text to highlight the novel contributions of our work:

"Our analysis provided a first assessment of Amazon ET representation in the CMIP6 climate models, showing they struggled to capture major features of Amazon ET, including spatial and seasonal variability across the Amazon basin. Furthermore,

CMIP6, which represents the latest generation of coupled climate models, showed little evidence of improvement in the representation of Amazon ET compared to CMIP5, highlighting the need for further process-based model development."

Our analysis focused on long-term monthly data. Future work could analyse monthly data in more detail, as suggested by the referee.

Other changes

We noticed that our Amazon LAI values were implausibly low (Amazon mean LAI value of 3.6 m2/m2), likely due to inadequate quality control during data processing. We have changed to use a quality-controlled MODIS MOD15A2H Collection 6 LAI dataset provided by Boston University (Amazon mean LAI value of 4.4 m2/m2). The main difference to the results arising from this change is that catchment-balance ET is no longer well related to spatial variation in LAI. The new figure and paragraph describing these results are copied below (see attached Fig. 2). There were no meaningful changes to any of the rest of the results.

"To understand the drivers of spatial variation in Amazon ET, we compared catchment-scale estimates against catchment-means of precipitation, surface radiation and LAI (Fig. 4). Since there were only eleven data points in the analysis (representing the Amazon and ten sub-catchments), statistical power was relatively low. However, we found spatial variation in catchment-balance ET showed some indication of an influence from radiation (r=0.38, p=0.25, Fig. 4h), but not precipitation (r=0.14, p=0.68, Fig. 4a) or LAI (r=0.06, p=0.87, Fig. 4o). This result tentatively suggests that spatial variation in radiation explains more of the spatial variability in ET across Amazon sub-catchments than other variables. None of the ET products and models analysed captured positive relationships between catchment-mean ET and radiation. ET from ERA5 and the CMIP ensembles instead showed negative associations with radiation (Fig. 4l–n), and, along with GLEAM ET, positive relationships with precipitation (Fig. 4d–g), indicative of water availability influencing spatial variation in ET (Fig. 4d–g).

These results confirm that the reanalysis and climate models analysed here struggled to capture spatial patterns in Amazon ET due to misrepresentation of the controlling drivers, specifically the relative importance of precipitation and net radiation. ET from ERA5 and the models also showed positive correlations between LAI and ET (Fig. 4s–u), not seen in the satellite observations. However, it should be noted that satellite LAI was generally lower and showed less spatial variability than other LAI datasets over the Amazon (Fig. S8i–l), likely due to the satellite sensor being insensitive to variation in LAI over areas of dense tropical forest (Myneni et al., 2002, Yan et al., 2016a). This could hamper our ability to accurately assess the extent to which LAI influences spatial variation in ET."

References

Paredes-Trejo, F. J., Barbosa, H. A. & Lakshmi Kumar, T. V. 2017.   Validating CHIRPS-based satellite precipitation estimates in Northeast Brazil.  Journal of Arid Environments, 139, 26-40.

Please also note the supplement to this comment:
https://hess.copernicus.org/preprints/hess-2020-523/hess-2020-523-AC3-supplement.pdf

———————————————

**Fig. 1.** Interannual variation in evapotranspiration from 2001 to 2019.

[Figure]

**Fig. 2.** Controls on spatial variation in Amazon evapotranspiration.

---

## Author Response (AR1)

**Response to Editor's comments**

**Dear Editor,**

Many thanks for reviewing our manuscript and ensuring our work reaches the highest possible standard. We have updated the manuscript in response to your comments, and have made the data and code available in appropriate repositories. Further details are provided in the text below.

We hope that our work will be considered suitable for publication in HESS, and we look forward to hearing from you soon.

Best wishes,

Jess Baker

**Dear authors,**

Thank you for your detailed responses to the referee comments, which were very supportive, but also suggested a few improvements and pointed out potential deficiencies that needed to be addressed. I think that you addressed most of the points very well and that your proposed revisions will make the paper a very valuable addition to the scientific literature. In addition to your proposed edits, I would also like you to consider the following points:

- Referee #3 was missing a clear statement "in what ways the current paper adds up to the currently reported literature on uncertainties in ET products (e.g. Mueller et al 2014) or in other words what is new here that we didn't know before?". In your response to the referee's comment about Line 555-560, you do highlight the novel aspects of your analysis, but I believe that it would be good to discuss more systematically in the introduction already what has been done to date (e.g. Mueller and Seneviratne 2014 and citations therein) and what gaps the present study intends to fill, which, as I understand it, goes beyond the application of old methods for bias detection to the new CMIP6 data, as mentioned in Line 130. In particular, the comparison with the catchment water balance and the associated uncertainty bounds are an important addition in my opinion. The paragraph starting in Line 123 could be a good place for this.

Thanks for raising this point. We had indeed not fully emphasised that the novelty of our approach comes from benchmarking the CMIP model output and remote sensing datasets against our catchment-balance estimates of ET and its associated uncertainty. As suggested, we have expanded the introduction to highlight this point.

"Finally, representation of Amazon ET in coupled climate models is still underdeveloped, in part due to limited high-quality reference observations. To overcome uncertainties in benchmarking data, Mueller and Seneviratne (2014) utilised a synthesis of 40 observational, reanalysis and land-surface model datasets (Mueller et al., 2013) to evaluate 14 models from the fifth Coupled Model Inter-comparison Project (CMIP5). Their analysis showed that Amazon ET tended to be overestimated at the annual scale, but underestimated from June to August. More recently it was observed that 28 out of 40 CMIP5 models misrepresented the controls on Amazon ET, with implications for future precipitation projections in the region (Baker et al., in review). Other assessments of CMIP5 models over the Amazon have found that choice of reference ET dataset can have a large impact on model performance metrics (Schwalm et al., 2013, Baker et al., 2021). Catchment-balance analysis accounting for changes in groundwater storage, offers an alternative approach for directly quantifying Amazon ET and its associated uncertainty at the monthly timescale, but to our knowledge has not previously been applied to evaluate climate models. With output from the sixth generation of CMIP models now available (Eyring et al., 2016), there is an opportunity to extend earlier evaluation studies by comparing simulated Amazon ET against catchment-balance estimates, and thus provide a first assessment of model performance over the Amazon."

- Data and code availability: Thank you for providing links to the datasets used, a summary of the

extracted data on zenodo and the scripts for raw data processing on github. Could you provide the full link to the zenodo repository, instead of just the doi? This would make it easier to find: <a href="https://doi.org/10.5281/zenodo.4271331">https://doi.org/10.5281/zenodo.4271331</a>

Good idea – we have now provided the link to the repository.

Since github.com is not a suitable permanent repository as outlined in the HESS data policy, would you be able to publish the relevant version of your github repo on zenodo as well? Zenodo offers a very convenient way to link up to a github repo.

We have created a 'release' of the github repository, which can now be obtained from Zenodo (https://zenodo.org/record/4580447#.YEC wi2l1hE).

The data provided in zenodo only contains basin-mean monthly estimates, but Fig. 3 and Fig. 6 show sub-basins. The uncertainty bounds in Fig. 6 are also not included in the zenodo file and it is not clear how the numbers in zenodo actually connect to the original datasets. Would you be able to describe how this table can be recreated from the original datasets using the scripts provided, and similarly, how the sub-basin data and error bounds can be recreated? This would help tremendously, should a reader wish to do a similar analysis for different datasets, as they would be able to verify that they are actually doing the same thing. You stated that the python scripts used for data analysis are available from the authors upon request. Is there a reason not to add them to the scripts used for raw data processing? I thank you already for your additional efforts to make your data analysis FAIR, as described in the HESS data policy (https://www.hydrology-and-earth-system-sciences.net/policies/data\_policy.html).

We have now uploaded all of the data analysis scripts, including the script to calculate catchment balance ET for the Amazon and its sub-basins, and scripts to extract basin-mean values from gridded netcdf files, in addition to the data processing scripts previously uploaded to github. The readme explains what each script does, and how to generate the table of catchment-scale estimates of Amazon evapotranspiration.

We also provide a script to estimate the errors in the catchment-balance ET for the Amazon. These errors are included as a separate table, accessible here: https://zenodo.org/record/4580292#.YECz-i2l23c.

For the sub-basins, we only used climatological mean ET. These values can be obtained using the provided script 'get\_catchment\_balance\_ET\_all\_basins.py'.

---

## Author Response (AR2)

Dear Editor,

We have made the final technical corrections to the manuscript and supplementary information. Many thanks again for your work handling our paper. We have found the process of publishing in HESS a very positive experience.

Many thanks,

Jess Baker

**Editor Decision: Publish subject to technical corrections** (05 Mar 2021) by Stan Schymanski Comments to the Author: Dear authors,

Thank you very much for addressing all comments very thoroughly and further improving the manuscript. A special thanks for making available all the scripts used to process the original data. Could you add the link and/or doi of the HESSD manuscript to each of the zenodo records?

We have now added the link to the HESS paper to both Zenodo records.

Except for a couple of technical corrections listed below I see no more issues to be addressed before publication. I am confident that the readership of HESS will find your work very helpful.

Best regards, Stan

**TECHNICAL CORRECTIONS:**

I324: ...due to there being...
Corrected.
I658: ...such as sap-flow.
Corrected.
Table S2: Could you explain the variables in the table caption, i.e. units and references to the equations used to compute them?

The caption for Table S2 is now as follows:

**Table S2 – Seasonal variation in Amazon catchment-balance error estimates.** Absolute uncertainties (in mm) in precipitation ( $\sigma_P$ ), river runoff ( $\sigma_R$ ), change in groundwater storage ( $\sigma_{dS}$ ), and evapotranspiration ( $\sigma_{ET}$ ), and the relative uncertainty (in %) in evapotranspiration ( $\upsilon_{ET}$ ).  $\sigma_P$  was estimated as the random error ( $\sigma_{P\_random}$ ) plus the systematic error ( $\sigma_{P\_bias}$ ), combined in quadrature.  $\sigma_{P\_random}$  was calculated following Eq. (4), from Huffman (1997):  $\sigma_{P\_random} = \overline{r} \left[ \frac{H-p}{p_N} \right]^{\frac{1}{2}}$  where  $\overline{r}$  is the climatological mean precipitation over the basin, H is a constant (1.5), p is the frequency of non-zero rainfall and N is the number of independent precipitation samples (defined as the number of Amazon pixels with finite P measurements in each month). For  $\sigma_{P\_bias}$ , we used the value of – 3.6 % from Table 4 in Paredes-Trejo et al. (2017).  $\sigma_R$  was estimated as 5% of monthly river flow (Dingman, 2015). Uncertainty in groundwater storage was quantified by combining GRACE measurement errors and leakage errors in quadrature. For these, we used Amazon-specific values from the literature (6.1 and 0.9 mm for measurement and leakage errors, see Table 1 in Wiese et al. (2016)). Since  $\frac{dS}{dt}$  values were calculated using data from two consecutive months, groundwater error values were multiplied by  $\sqrt{2}$  to obtain  $\sigma_{dS}$  (e.g. Maeda et al., 2017).  $\sigma_{ET}$  was estimated using  $\sigma_{ET} = \sqrt{\sigma_P^2 + \sigma_R^2 + \sigma_{dS}^2}$ , and  $\upsilon_{ET} = \frac{\sigma_{ET}}{ET} \times 100$ . For further details please see the mean mean The Amazon particle of the relative for this graph as a considered for this graph as a considered for this graph as a constant is indicated by by by batching in Figure 1.

the main paper. The Amazon region considered for this analysis is indicated by blue hatching in Figure 1.

Finally, I changed the shading in the inset map in Figure 6 from grey shading to hatched black lines to match the inset map in the other figures.